# Non-Excitatory Amino Acids, Melatonin, and Free Radicals: Examining the Role in Stroke and Aging

**DOI:** 10.3390/antiox12101844

**Published:** 2023-10-10

**Authors:** Victoria Jiménez Carretero, Eva Ramos, Pedro Segura-Chama, Adan Hernández, Andrés M Baraibar, Iris Álvarez-Merz, Francisco López Muñoz, Javier Egea, José M. Solís, Alejandro Romero, Jesús M. Hernández-Guijo

**Affiliations:** 1Department of Pharmacology and Therapeutic, Teófilo Hernando Institute, Faculty of Medicine, Universidad Autónoma de Madrid, Av. Arzobispo Morcillo 4, 28029 Madrid, Spain; victoria.jimenez@uam.es (V.J.C.); iris.alvarez.merz@hhu.de (I.Á.-M.); 2Department of Pharmacology and Toxicology, Faculty of Veterinary Medicine, Complutense University of Madrid, 28040 Madrid, Spain; eva.ramos@ucm.es (E.R.); manarome@ucm.es (A.R.); 3Investigador por México-CONAHCYT, Instituto Nacional de Psiquiatría “Ramón de la Fuente Muñiz”, Calzada México-Xochimilco 101, Huipulco, Tlalpan, Mexico City 14370, Mexico; segurapd@gmail.com; 4Institute of Neurobiology, Universidad Nacional Autónoma of México, Juriquilla, Santiago de Querétaro 76230, Querétaro, Mexico; adanherdez@gmail.com; 5Department of Neurosciences, Universidad del País Vasco UPV/EHU, Achucarro Basque Center for Neuroscience, Barrio Sarriena, s/n, 48940 Leioa, Spain; andresmateo.baraibar@ehu.eus; 6Faculty of Health Sciences, University Camilo José Cela, C/Castillo de Alarcón 49, Villanueva de la Cañada, 28692 Madrid, Spain; flopez@ucjc.edu; 7Neuropsychopharmacology Unit, Hospital 12 de Octubre Research Institute (i + 12), Avda. Córdoba, s/n, 28041 Madrid, Spain; 8Molecular Neuroinflammation and Neuronal Plasticity Research Laboratory, Hospital Universitario Santa Cristina, Health Research Institute, Hospital Universitario de la Princesa, 28006 Madrid, Spain; javier.egea@inv.uam.es; 9Neurobiology-Research Service, Hospital Ramón y Cajal, Carretera de Colmenar Viejo, Km. 9, 28029 Madrid, Spain; jota.m.solis@gmail.com; 10Ramón y Cajal Institute for Health Research (IRYCIS), Hospital Ramón y Cajal, Carretera de Colmenar Viejo, Km. 9, 28029 Madrid, Spain

**Keywords:** melatonin, non-excitatory amino acids, free radicals, stroke, aging

## Abstract

The aim of this review is to explore the relationship between melatonin, free radicals, and non-excitatory amino acids, and their role in stroke and aging. Melatonin has garnered significant attention in recent years due to its diverse physiological functions and potential therapeutic benefits by reducing oxidative stress, inflammation, and apoptosis. Melatonin has been found to mitigate ischemic brain damage caused by stroke. By scavenging free radicals and reducing oxidative damage, melatonin may help slow down the aging process and protect against age-related cognitive decline. Additionally, non-excitatory amino acids have been shown to possess neuroprotective properties, including antioxidant and anti-inflammatory in stroke and aging-related conditions. They can attenuate oxidative stress, modulate calcium homeostasis, and inhibit apoptosis, thereby safeguarding neurons against damage induced by stroke and aging processes. The intracellular accumulation of certain non-excitatory amino acids could promote harmful effects during hypoxia-ischemia episodes and thus, the blockade of the amino acid transporters involved in the process could be an alternative therapeutic strategy to reduce ischemic damage. On the other hand, the accumulation of free radicals, specifically mitochondrial reactive oxygen and nitrogen species, accelerates cellular senescence and contributes to age-related decline. Recent research suggests a complex interplay between melatonin, free radicals, and non-excitatory amino acids in stroke and aging. The neuroprotective actions of melatonin and non-excitatory amino acids converge on multiple pathways, including the regulation of calcium homeostasis, modulation of apoptosis, and reduction of inflammation. These mechanisms collectively contribute to the preservation of neuronal integrity and functions, making them promising targets for therapeutic interventions in stroke and age-related disorders.

## 1. Introduction

Melatonin (*N*-acetyl-5-methoxytryptamine), an indoleamine secreted from the pineal gland that plays a crucial role in regulating circadian rhythms and sleep–wake cycles [1,2], has garnered significant attention in recent years due to its diverse physiological functions and potential therapeutic benefits. Emerging research suggests that melatonin may have broader implications beyond its role in sleep [3,4,5,6,7,8,9,10,11,12,13,14,15]. In this context, the aim of this review is to explore the intricate relationship between melatonin, free radicals, and non-excitatory amino acids, and their role in stroke and aging. By understanding these mechanisms, we can gain insights into potential interventions for age-related diseases and stroke prevention. The aim of this review is to explore the potential role of melatonin in stroke and aging, shedding light on its neuroprotective effects and potential therapeutic applications.

Non-excitatory amino acids, such as glycine and taurine, have been shown to possess neuroprotective properties in stroke and aging-related conditions [14,16,17]. Glycine acts as a co-agonist at the N-methyl-D-aspartate (NMDA) receptor, thereby counteracting the excitotoxicity caused by excessive glutamate release during stroke. Excitotoxicity leads to neuronal cell death and contributes to the progression of neurodegenerative diseases. Taurine, on the other hand, exhibits multiple neuroprotective effects, including antioxidant and anti-inflammatory properties [18]. It can attenuate oxidative stress, modulate calcium homeostasis, and inhibit apoptosis, thereby safeguarding neurons against damage induced by stroke and aging processes [19]. Furthermore, the intracellular accumulation of certain non-excitatory amino acids, which were previously not considered excitotoxic, could promote harmful effects during hypoxia-ischemia episodes through the activation of NMDAR [20,21]. Therefore, given that NMDAR antagonists have been shown to be ineffective in clinical trials of ischemic stroke [22], blocking the amino acid transporters involved in the process could be an alternative therapeutic strategy to reduce ischemic damage.

Studies have shown that melatonin exhibits neuroprotective properties in the context of stroke [23]. By reducing oxidative stress, inflammation, and apoptosis, melatonin has been found to mitigate ischemic brain damage caused by stroke [24]. Additionally, melatonin’s ability to regulate cerebral blood flow and attenuate excitotoxicity further contributes to its neuroprotective effects [25]. These findings suggest that melatonin may hold promise as an adjunct therapy for stroke patients.

Over 75% of strokes are caused by a thromboembolic mechanism. Of the remaining cases, after extended diagnoses, it turns out that half are also caused by this mechanism. The current treatment for the prevention of the development of thrombi is the use of Direct Acting Oral Anticoagulants (DOACs) which have a specific mechanism of action that allows them to interfere with the blood coagulation process directly and more selectively than anticoagulants, i.e., traditional drugs such as warfarin [26,27]. DOACs act on key proteins involved in the coagulation cascade, specifically coagulation factors, to prevent clot formation: (i) they include drugs such as rivaroxaban, apixaban, and edoxaban whose mechanism of action is based on the inhibition of factor Xa, a central component in the coagulation cascade necessary to convert prothrombin to thrombin, an enzyme essential in clot formation; and (ii) drugs such as dabigatran, a direct thrombin inhibitor. Thrombin is a key enzyme in the coagulation cascade that converts fibrinogen to fibrin, a protein that forms the main structure of a clot. By inhibiting thrombin, dabigatran reduces the blood’s ability to form clots. However, it is important to note that these types of drugs can have interactions with melatonin. Current evidence suggests that melatonin inhibits platelet aggregation and might affect the coagulation cascade, altering fibrin clot structure and/or resistance to fibrinolysis [28,29,30,31]. The mechanisms behind melatonin-associated reduction in procoagulant response are not fully known.

Aging is associated, among other factors, with a decline in melatonin production, which may contribute to age-related disorders [32]. Melatonin’s antioxidant properties are particularly relevant in the context of aging, as oxidative stress plays a significant role in age-related neurodegenerative diseases such as Alzheimer’s and Parkinson’s [33]. By scavenging free radicals and reducing oxidative damage, melatonin may help slow down the aging process and protect against age-related cognitive decline. The impact of free radicals on aging has been a subject of investigation for many researchers. Several studies [34,35,36,37,38] revealed that the accumulation of free radicals, specifically mitochondrial reactive oxygen and nitrogen species (RONS), highly reactive chemicals, accelerates cellular senescence and contributes to age-related decline.

Furthermore, [39] explored the interplay between free radicals, inflammation, and aging, highlighting the role of oxidative stress in the aging process.

Oxidative stress arises from an imbalance between the production of RONS and the body’s ability to neutralize them. Free radicals, such as RONS, can cause cellular damage, leading to various diseases, including stroke and age-related disorders. Several studies have demonstrated the involvement of free radicals in the pathophysiology of stroke. Many reports have focused on how ROS contribute to oxidative stress and neuronal damage following an ischemic stroke [40,41,42]. Several comprehensive reviews of the role of free radicals in stroke have been published [43,44]. In this regard, melatonin has strong antioxidant activity, and it acts as a free radical scavenger by directly neutralizing RONS, thereby reducing oxidative damage [45]. Additionally, it enhances the activity of antioxidant enzymes and upregulates the expression of endogenous antioxidant molecules [45]. These actions collectively help mitigate the detrimental effects of free radicals and protect cells from oxidative, stress-induced injury.

Recent research suggests a complex interplay between melatonin, free radicals, and non-excitatory amino acids in stroke and aging. Melatonin’s antioxidant properties contribute to the reduction of free radicals, including RONS, thereby indirectly influencing the modulation of non-excitatory amino acids. Furthermore, melatonin has been shown to enhance the endogenous production of glycine and taurine, potentially amplifying their neuroprotective effects [46]. The neuroprotective actions of melatonin, glycine, and taurine converge on multiple pathways, including the regulation of calcium homeostasis, modulation of apoptosis, and reduction of inflammation. These mechanisms collectively contribute to the preservation of neuronal integrity and function, making them promising targets for therapeutic interventions in stroke and age-related disorders.

In summary, melatonin, free radicals, and non-excitatory amino acids play integral roles in stroke and aging processes. Melatonin’s potent antioxidant properties help mitigate oxidative stress, while non-excitatory amino acids provide neuroprotection by modulating excitotoxicity, calcium homeostasis, and inflammation.

### Mechanisms Underlying Local Blood Flow

The relationship between local blood flow and aging is a complex and multifaceted topic [47,48,49,50,51,52]. As we age, a number of changes occur in the cardiovascular system and blood vessels that can affect local blood flow in several ways: (i) with aging, arteries tend to become stiffer due to plaque buildup, atherosclerotic disease, and increased stiffness of arterial walls. This stiffness can decrease the arteries’ ability to dilate and contract properly, negatively affecting local blood flow; (ii) over time, capillary density may decrease, which may reduce the ability to deliver blood and nutrients to tissues efficiently; (iii) the endothelium is the inner layer of the arteries and plays a crucial role in regulating blood flow by releasing nitric oxide and other substances, but with age, endothelial function can deteriorate, which can affect the ability of the blood vessels to dilate and contract appropriately; (iv) aging is associated with an increase in the production of free radicals and oxidative stress, which can damage cells and blood vessels, contributing to vascular dysfunction and impairment of local blood flow; and (v) in some cases, aging can lead to reduced blood flow in specific tissues, which can contribute to age-related diseases, such as cardiovascular disease or Alzheimer’s disease [53,54,55,56].

Local blood flow is a fundamental process for the delivery of oxygen and nutrients to tissues, and the removal of waste products. Local blood flow is regulated by several mechanisms, including the action of substances such as nitric oxide (NO) and carbon monoxide (CO) [57,58,59,60]. Nitric oxide is a gas molecule produced by the endothelial cells that line the inside of blood vessels, especially arteries, and acts as a powerful vasodilator, relaxing the smooth muscles surrounding the arteries, allowing them to widen and increase local blood flow. This dilation of the vessels is essential to regulate blood flow and blood pressure in different parts of the body [61]. Although carbon monoxide is primarily known as a toxic gas when inhaled in large quantities, it is also produced naturally in the body as a result of the breakdown of hemoglobin [62]. In the context of local blood flow, it has been found that CO can act as a vasodilator similar to NO under certain circumstances [63,64,65,66,67,68]. CO can relax the smooth muscles of the arteries and increase local blood flow, although its role in this regulation is less known and less studied than NO. Both NO and CO play crucial roles in regulating local blood flow [69]. Its function is complex and is influenced by numerous factors, including tissue metabolic activity, blood pressure, the presence of other chemicals, and the health of the vascular endothelium. Dysfunction in the production or action of these molecules can contribute to vascular disorders [70], neurodegenerative pathologies [71], and others [72,73,74].

The guanylate cyclase pathway is related to the production of NO, which, among other physiological functions, plays an important role in the regulation of blood flow [75]. Melatonin is a hormone that regulates the circadian rhythm and plays a role in sleep and wakefulness. These two molecules are indirectly related through their effects on the vascular system. Melatonin can influence the guanylate cyclase pathway and NO production in the following way: (i) by an indirect route, since melatonin can have indirect effects on the vascular system through its influence on the autonomic nervous system and the regulation of vascular tone influencing the production of NO; and (ii) by a direct route, since on the one hand it has been described that melatonin causes an enhancement of the activity of the guanylate cyclase-cyclic GMP system [76] and NO interacts with melatonin as a long-range signaling molecule, and helps regulate oxidative homeostasis [77]; however, on the other hand it has been reported, and seems to be the most accepted by the scientific community, that when endogenous melatonin levels are elevated, it results in a significant decrease in NOS activity [78] via complex formation with calmodulin [79,80], and even melatonin synthetic analogs such as nitric oxide synthase inhibitors have been developed [81]. In summary, melatonin is known to modulate cGMP concentration via the MT2 receptor. It has been found to affect vascular function and blood pressure regulation, which involves cGMP signaling pathways in blood vessels. In this sense, melatonin regulates coronary vasomotor tone through the MT2 receptor and stimulation of PDE5, which in turn, increases degradation of cGMP. However, the relationship between melatonin and guanylate cyclase is complex and may vary depending on the specific tissues and conditions being studied [82,83,84].

Vascular smooth muscle contraction plays a relevant role in aging and disease. The processes that regulate vascular smooth muscle function, and thus, influence the vascular diameter, involve complex-interacting systems such as the renin–angiotensin–aldosterone system, sympathetic nervous system, immune activation, and oxidative stress [85,86,87,88]. Vascular smooth muscle contraction is triggered by an increase in intracellular free calcium concentration ([Ca^2+^]_i_), promoting actin–myosin cross-bridge formation. This contraction is also regulated by calcium-independent mechanisms involving RhoA-Rho kinase [89,90,91], protein Kinase C [88,92], and mitogen-activated protein kinase signaling [93,94], reactive oxygen species [95,96,97], and reorganization of the actin cytoskeleton [98,99,100]. Perturbations in vascular smooth muscle cell signaling and altered function influence vascular reactivity and tone, which are important determinants of vascular resistance and blood flow [101]. The regulation of vascular diameter and consequently vascular resistance depends on the activation status of the contractile machinery involving actin: myosin interaction in vascular smooth muscle cells [102]. Changes in [Ca^2+^]_i_, ion fluxes, and membrane potential lead to calcium–calmodulin-mediated phosphorylation of the regulatory myosin light chains and actin–myosin cross-bridge cycling with consequent rapid vasoconstriction [103]. Calcium-independent mechanisms associated with altered calcium sensitization and actin filament remodeling and increased bioavailability of reactive oxygen species (ROS) (oxidative stress), also modulate vascular contraction [104].

In any case, melatonin is capable of interfering with all of these agents. Melatonin MT1 and MT2 receptors are G-protein-coupled receptors that are expressed in various parts of the CNS and peripheral organs (blood vessels included). Melatonin receptors mediate a plethora of intracellular effects depending on the cellular milieu. These effects comprise changes in intracellular cyclic nucleotides (cAMP and cGMP) and calcium levels and activation of protein kinase C [105,106,107]. Melatonin attenuated choroidal neovascularization is an important characteristic of advanced wet age-related macular degeneration and leads to severe visual impairment among elderly patients, reduced vascular leakage, and inhibited vascular proliferation via inhibition of the RhoA signaling pathway [108]. Melatonin treatment restored impaired contractility via the normalization of Ca^2+^ handling and Ca^2+^ sensitization pathways [109]. Melatonin exerts a modulation of the mitogen-activated protein kinases mediating intracellular processes [110,111], and so, leads to a protective effect during hepatic ischemia-reperfusion injury [112] and attenuates cerebral ischemic injury [113].

Mastoparan-7, an analog of the peptide mastoparan, which is derived from wasp venom, is a direct activator of Pertussis toxin-sensitive G-proteins that produce several biological effects in different cell types [114,115]. Mastoparan-7 activates guanine nucleotide-binding proteins (G-proteins), increases cytoplasmic calcium concentration, and induces smooth muscle contraction [116,117,118]. Considering that the receptors for mastoparan and melatonin activate Gi, they may share the same pool of inhibitory G-proteins in a similar way to what happens with opioidergic and purinergic receptors [119,120], or the activation of both receptors may even have a synergistic effect. In any case, it is necessary to carry out studies to understand the interaction at the vascular level between these two molecules. On the other hand, it has been reported that mastoparan induces the production of ROS [121] via arachidonic cascade [122], an important effect considering that precise morphogenetic and cellular mechanisms act in endothelial cells to drive angiogenesis during growth and throughout adulthood, and that ROS and their metabolism are proving to be crucial participants in the shaping and stabilizing of blood vessels [123]. In fact, ROS derived from NADPH oxidase as well as mitochondria play an important role in promoting the angiogenic switch from quiescent endothelial cells [124]; often, the same mechanisms are responsible for the insurgence of vascular-associated pathologies, the excessive ROS generation results in the initiation and progression of cardiovascular diseases [125].

The reactivity of small resistance vessels can be influenced by various factors, including the nature of tissue response, and it is indeed possible for vasoconstrictors to have a direct effect on these vessels, as well as influence their behavior through the production of free radicals [126,127]. Small resistance vessels, arterioles, play a crucial role in regulating blood flow and blood pressure. Their reactivity refers to their ability to constrict or dilate in response to various physiological and pharmacological stimuli. Vasoconstrictors cause blood vessels to constrict, leading to an increase in vascular resistance and blood pressure. Some vasoconstrictors, such as mastoparan-7, can have a direct effect on small resistance vessels by binding to receptors on the vessel walls and causing them to contract. On the other hand, free radicals can lead to damage to cells and tissues, including endothelial cells. Some vasoconstrictors, particularly when present in high concentrations or under certain conditions, can stimulate the production of free radicals. These free radicals can then impair the function of endothelial cells and contribute to vasoconstriction. In summary, the reactivity of small resistance vessels can be influenced by vasoconstrictors both through their direct action on vessel smooth muscle and through their ability to promote the production of free radicals, which can have detrimental effects on the vessels’ function. This interplay between vasoconstrictors and free radicals can have implications for blood pressure regulation and overall cardiovascular health.

Thus, the general objective of this review focuses on evaluating the role of melatonin, free radicals, and non-excitatory amino acids in stroke and aging, focusing on the multiple properties of melatonin, such as antioxidants, and neuroprotection. provided by non-excitatory amino acids in the modulation of excitotoxicity and calcium homeostasis, to understand its therapeutic potential in the prevention and treatment of stroke. From this general objective, we can define several specific objectives to guide the review: (i) to analyse the current scientific evidence on the role of melatonin in mitigating oxidative stress in the context of stroke and aging; (ii) to examine the molecular mechanisms involved in the antioxidant properties of melatonin and how these may influence the reduction in brain damage associated with stroke; (iii) to examine the properties of melatonin linked to inflammatory processes and evaluate its involvement in the damage associated with stroke; (iv) to investigate the effects of free radicals and oxidative stress on the development and progression of strokes and how melatonin can counteract these processes; (v) to review the scientific literature related to non-excitatory amino acids and their ability to provide neuroprotection by modulating excitotoxicity and calcium homeostasis in the context of stroke; (vi) to identify possible interaction pathways between melatonin, non-excitatory amino acids, and other biological systems relevant to the pathogenesis of stroke; (vii) to evaluate the effectiveness and potential therapeutic benefits of the administration of melatonin and non-excitatory amino acids in preclinical models and clinical trials related to stroke; and (viii) to synthesize the findings of the review to provide an overview of how melatonin and non-excitatory amino acids can be considered as potential therapeutic strategies in the prevention and treatment of stroke.

## 2. Stroke and Aging

Stroke is the most common cerebrovascular disease, being the second largest cause of death and the leading cause of disability worldwide [128,129]. From 1990 to 2019, the number of strokes worldwide has increased by 70%, with an incidence of around 15 million new strokes globally [130].

Although there is a small genetic predisposition to suffer from a stroke, the majority of them are related to other factors [131]. Eighty-seven percent of strokes occur in individuals aged above 49 years, indicating a relationship between the increase in life expectancy and the prevalence of strokes. The risk of stroke increases with each passing year of age [132,133]. Additionally, the prognosis for outcomes after a stroke is worse for elderly individuals, with women experiencing a higher mortality rate than men [134,135].

Among the modifiable risk factors for stroke are hypertension, hyperlipidemia, diabetes mellitus, smoking, physical inactivity, poor diet, and obesity [136]. However, many risk factors are non-modifiable, with advanced age being the most significant risk factor. Like other diseases associated with aging, there are different mechanisms that play a role in developing a stroke, acting additively or synergistically. Some of these mechanisms include genomic instability, telomere attrition, epigenetic alterations, loss of proteostasis, deregulated nutrient sensing, stem cell exhaustion, altered intercellular communication, cellular senescence, mitochondrial dysfunction, disabled macroautophagy, chronic inflammation, and dysbiosis [137,138].

Strokes can be either hemorrhagic or ischemic. The former occurs because of a rupture of a blood vessel in the brain, causing bleeding. On the other hand, the latter, which is the most prevalent form, is caused when the blood supply to a certain region of the brain is reduced due to an obstruction in a blood vessel [139]. This blood supply impairment leads to a reduction in the delivery of oxygen and glucose, resulting in diminished energetics required to maintain ionic gradients. Consequently, it causes a loss of membrane potential, depolarization of neurons and glial cells, and an increased release of excitatory neurotransmitters (such as glutamate) that ultimately lead to neuronal death [140]. The excessive release of glutamate activates post-synaptic NMDAR and metabotropic glutamate-receptors, as well as L-type voltage-gated calcium channels. Their activation contributes to a Ca^2+^ overload in the cells, activating intracellular signaling cascades that ultimately result in an increased generation of RONS [43,129,140].

The aging process also causes structural and functional impairments in the neurovascular unit (NVU), which is composed of neurons, astrocytes, endothelial cells of the blood-brain barrier (BBB), myocytes, pericytes, and extracellular matrix components [134,141]. The vulnerability of this NVU exacerbates the risk and severity of ischemic stroke by aggravating its initial phases and reducing the subsequent tissue repair and regeneration processes [134].

Epigenetic modifications correlate with chronological age, but since they are influenced by environmental factors, they can accelerate biological aging [142]. This biological age is the most determinant of the two when it comes to determining the outcome of an ischemic stroke [143]. Impairment of macroautophagy is directly connected with oxidative stress and inflammation [144], and dysbiosis is closely related to neuroinflammation and stroke outcomes [145].

Chronic inflammation associated with aging is a result of cell damage that accumulates with age, senescent cells, dysbiosis, immunosenescence, and the increasing activation of the coagulation system [146]. In the brain, chronic inflammation leads to an exacerbated microglial inflammatory response to stroke, as well as overall inflammation that persists longer than usual after the disease due to elevated expression of pro-inflammatory molecules [134,147].

## 3. Non-Excitatory Amino Acids as an Alternative Therapeutic Strategy to Reduce Ischemic Damage

The majority of excitatory and inhibitory synapses in the brain utilizes, as neurotransmitters, the amino acids glutamate [148] and GABA [149], respectively. Thus, no wonder that both amino acids are involved directly or indirectly in the plethora of brain disorders, which, in the case of glutamate, give rise to the hypothesis of “excitotoxicity” [150], ascribing this neurotransmitter as the main culprit causing cell damage. Experimental evidences obtained on this topic created an early enthusiasm that was cooled down when clinical trial evidence came to show the inefficiency of NMDAR antagonists to protect cell damage developed during ischemia [22]. Nowadays, the neuroprotective strategies against ischemic excitotoxicity have several targets including extracellular glutamate levels and its receptors and the downstream activation of cell death pathways [151].

Various brain pathologies are associated with swelling of both neurons and astrocytes, which activates volume-regulatory mechanisms that involve the efflux of osmolytes including AA such as glutamate, aspartate, alanine, glutamine, glycine, and taurine, whose efflux through volume-regulated anion channels (VRAC) [152,153,154] try to recover cellular volume even if the cause of the swelling is still present [152,155]. Furthermore, the loss of plasma membrane integrity produced during ischemic cell death not only releases glutamate but many cytoplasmic substances, among them other non-excitatory amino acids not directly involved in neurotransmission. In this sense, it is important to highlight the observation showing that the ischemic damage volume was proportional to the amount of amino acids released during ischemia, and that it was not exclusively neurotoxic [156]. In relation with this issue, we have identified a group of non-excitatory amino acids, L- and D-alanine, L- and D-serine, L- and D-threonine, L-glutamine, glycine, L-histidine, and taurine, whose individual application to rat hippocampal slices at high unphysiological concentrations (10 mM) produces extracellular space shrinkage, which is probably due to the intracellular accumulation of the applied amino acids [157]. This was not produced by any amino acid applied at high concentrations, because a 10 mM concentration of either L-arginine, L-leucine, L-methionine, L- or D-proline, or L-valine did not induce changes in evoked field potentials compatible with cell swelling [157] (Figure 1).

In a later study, we observed that the application of a mixture of L-alanine, L-glutamine, glycine, L-histidine, L-serine, taurine, and L-threonine at their plasmatic concentrations also induced the intracellular accumulation of these amino acids, accompanied by an increase in the slice electrical resistance, which indicates the occurrence of cellular swelling [20] (Figure 2). This phenomenon was not associated with changes in the basic electrical properties of recording neurons, and it was resistant to the presence of an NMDA receptor antagonist, suggesting that this AA mixture does not activate ionotropic glutamate receptors. Extracellular increments of these non-excitatory amino acids at such concentration ranges have been detected in experimental models of ischemia [158,159,160] and in patients with head injury [161], subarachnoid hemorrhage [161,162,163], and acute focal ischemia [164]. The impact of these amino acids on the evolution of ischemic-induced damage is unknown. Recent experiments carried out by our group have shed light on this question.

In rat hippocampal slices, we observed that the induction of 40 min of hypoxia caused the full loss of synaptic responses, evidenced by the disappearance of field EPSPs, which recovered completely after reoxygenation, as previously demonstrated by many groups [166,167]. In contrast, the presence of the seven-AA mixture during the hypoxia period made both the synaptic silencing [20] and the loss of membrane potential irreversible [165] (Figure 3), indicating that neurons have suffered irreversible damage. Lately, we showed that all the seven amino acids were not required to induce this detrimental effect, because with a mixture of just four AA (alanine, glutamine, glycine, and serine) the hypoxia effect was irreversible [21,165]. We also found that the presence of glutamine in this mixture was necessary but not sufficient (i.e., neither the application of alanine, glycine plus serine without glutamine, nor the individual application of glutamine during hypoxia induced permanent synaptic silencing). These results are remarkable given that L-glutamine is the most concentrated AA in the mixture and it is a substrate for the synthesis of glutamate and GABA [21]. Nevertheless, equimolar substitution of glutamine by histidine or threonine (a mixture of alanine, glycine and serine, plus histidine or threonine at 733 µM concentration) also produced detrimental effects of synaptic transmission [165]. Moreover, the individual application of each of the seven AA at a final concentration such as that of the whole AA mixture (2.1 mM) showed that serine, glycine, and histidine are the most potent AA, causing the irreversible loss of synaptic transmission during hypoxia, while alanine and glutamine, or taurine and threonine, allowed partial or total recovery of synaptic potentials, respectively [165]. This suggests that both the identity and quantity of each AA in a mixture determine the magnitude of hypoxic-induced synaptic transmission deterioration.

Recent studies, in which images of neurons and astrocytes in hippocampal slices were obtained with a multiphoton system, have clearly shown that the permanent synaptic loss induced by hypoxia in the presence of alanine, glutamine, glycine, and serine, was accompanied by the swelling of both neurons and astrocytes [21]. In the case of astrocytes, it was observed that they also swell in normoxic conditions with the mere presence of the AA mixture. Additionally, this work revealed the existence of irreversible dendritic beading and mitochondrial swelling, which demonstrates the acute damage caused under these experimental conditions.

Some of the detrimental effects caused by these non-excitatory amino acids during hypoxia, such as permanent synaptic silence, neuronal swelling and dendritic beading are attributable to the activation of NMDA receptors, because they were prevented by an NMDA antagonist [20,21]. However, this NMDA antagonist does not inhibit the astroglial swelling induced by the AA mixture and hypoxia [21], indicating that neuronal swelling and damage is secondary to the swelling occurring in astrocytes, which direct- or indirectly could favour the neurotoxic processes mediated by glutamate underlying ischemic damage. Glial swelling may “passively” increase extracellular glutamate concentrations, and those of other interstitial substances, just by the reduction of extracellular volume as a consequence cellular edema. Furthermore, glial swelling activates volume-regulated anion channels (VRAC) which operate as a glutamate efflux pathway [168,169]. The use of a VRAC antagonist has revealed that these channels seem to participate in most of the detrimental effects induced by the AA mixture during hypoxia, including synaptic silencing, neuronal and astroglial swelling, and dendritic beading [21].

The astrocytic swelling, that was observed when the AA mixture was applied either in normoxic or hypoxic conditions, as pointed out above, was resistant to the presence of an NMDA antagonist [21]. We initially hypothesized that the intracellular accumulation of the applied amino acids, driven by specific amino acid transporters, creates the osmotic conditions promoting astroglial swelling. Several systems of neutral amino acid transporters with a substantial brain expression, have the amino acids composing our AA mixtures as substrates. Among these transporters, we examine the participation of the sodium-dependent systems, A, L, N, and ASCT2 [170,171], in the deleterious effects provoked by the non-excitatory amino acid that we are describing. Based on substrate specificity of these transporters and the use of specific transporter inhibitors (MeAIB against system A [172], and BCH against system L [173,174]) we considered the participation of system A [20] and system L [165] in the deleterious effect of non-excitatory amino acid on synaptic recovery unlikely. In contrast, we have compelling evidence that system ASC was involved in this phenomenon. ASC transporters are sodium-dependent AA exchangers, primarily located in astrocytes and neurons [175,176]. The following evidence supports the contribution of the variant ASCT2 of this transport system in the harmful effects of AAs during hypoxia: (i) L-threonine is one of the preferred substrates of ASCT2, along with L-alanine, L-serine, and L-glutamine [171]; (ii) when L- or D-threonine replaces L-glutamine in the four-AA mixture, synaptic silencing occurs after hypoxia; and (iii) GPNA, an inhibitor of ASCT2 [177,178], affects the irreversible silencing of synaptic potentials [21,165] and neuronal and astroglial swelling [21] caused by alanine, glycine, glutamine, and serine in hypoxia. Another feature of ASCT2 relevant to our study is that it can release D-serine in exchange for L-alanine and L-glutamine [179]. This opens an alternative possibility where, during hypoxia, the AA mixture stimulates the release of D-serine from neuronal ASCT2 which would participate in the coactivation NMDAR involved in neuronal soma swelling and dendritic beading.

Because glutamine, alanine, serine, and histidine also activate system N [180,181], the question remains as to whether this transport system is involved in the deleterious process described here. When histidine replaces glutamine in the AA mixture also containing alanine, glycine, and serine, there was a permanent loss of synaptic potentials upon reoxygenation, and histidine was greatly accumulated in the slice [165], suggesting the involvement of system N in the detrimental effect of these amino acids.

Another aspect to consider is that the reduction of ATP levels induced by hypoxia [182] could decrease the activity of Na^+^/K^+^-ATPase, thereby reducing the driving force necessary for the active transport of AA. However, there is evidence showing that concentrative AA transporters continue to function during short or mild periods of ischemia [183,184], indicating that even under hypoxic conditions, some transporters can still accumulate AA.

In summary, the intracellular accumulation of certain AA, which are not considered excitotoxic, could promote harmful effects during hypoxia-ischemia episodes through the activation of NMDAR. These non-excitatory amino acids at plasma concentrations provoke cellular swelling during hypoxia [20,21], which can increase extracellular glutamate concentration in two ways: (i) releasing glutamate through volume-regulated anion channels (VRAC) activated by astroglial swelling, which was induced by carrier-mediated amino acid accumulation, and (ii) the reduction of interstitial volume consequent to cellular swelling would increase the concentration of extracellular molecules, including some neurotoxins such as glutamate and D-serine. Therefore, given that NMDAR antagonists have been shown to be ineffective in clinical trials of ischemic stroke [22], blocking the AA transporters involved in the process described here could be an alternative or adjuvant therapeutic strategy to reduce ischemic damage (see Figure 4).

## 4. Neuroprotective Effects of Melatonin in Ischemic Stroke

### 4.1. Melatonin’s Effects on Cerebral Edema

One of the features of ischemic stroke is the disruption of the BBB, which is caused by the degradation of tight junctions between cells and enhanced transport of endothelial vesicles. As a result, vasogenic edema appears and worsens the patient’s prognosis [185], contributing to the high mortality rate associated with ischemic stroke [186]. Unfortunately, there is currently no effective treatment available [187].

Due to its high lipophilicity and ability to cross the BBB and enter the brain parenchyma easily, melatonin has been the focus of numerous studies aiming to better understand its mechanism of action and test its potential use as a treatment for stroke-induced brain edema [188,189]. In a study conducted by Kondoh et al., rats were subjected to one hour of middle cerebral artery occlusion (MCAO) followed by reperfusion. The effects of oral administration of melatonin (6.0 mg/kg) prior to MCAO and one day after were tested, and it was observed that the total volume of cerebral edema was reduced by over 40% in the melatonin-treated animals compared to the control group [190]. This could be explained by the ability of melatonin to reduce the increase in BBB permeability when administered at the time of reperfusion [191].

Furthermore, melatonin has been shown to reduce the upregulated expression of aquaporin-4 in astrocytes, which occurs during ischemic stroke-induced hypoxia. This reduction helps decrease water uptake by astrocytes and the formation of brain edema [192]. In melatonin-treated rats after cerebral ischemia, the BBB was better preserved compared to control rats, likely due to the restoration of downregulated Na^+^/K^+^/ATPase activity, which is necessary to maintain cell membrane integrity [193].

Moreover, melatonin can reduce the increase in vascular permeability by acting on endothelial cells [194] and lowering the concentrations of vascular endothelial growth factor (VEGF) and nitric oxide (NO), two factors that are elevated during edema [195]. In addition to these factors, melatonin can also decrease the levels of matrix metalloproteinase 9 (MMP-9), a protein secreted by pericytes and induced by interleukin-1β (IL-1β), which contributes to BBB damage. Therefore, melatonin plays a protective role in maintaining BBB integrity [196].

### 4.2. Melatonin’s Effects on the Post-Stroke Inflammatory Response

In response to the necrotic cells that appear within the infarct area after an ischemic stroke, and because of an increase in oxidative stress, an inflammatory cascade is activated, in which danger-associated molecular pattern molecules (DAMPs) released from the necrotic cells and other immune molecules interact with TLR4 and other toll-like receptors, activating the microglia. This activation leads to increased cytokine production, damaging the BBB and promoting the migration of leukocytes to the ischemic injury. It also deregulates the expression of many adhesion molecules, such as integrins and selectins [197,198]. In this regard, melatonin has been shown to ameliorate inflammation in different tissues by downregulating the expression of NF-κB, a transcription factor that encodes various proinflammatory cytokines, through its action on different pathways (Figure 5) [199,200,201]. Additionally, it can modulate the activation of astrocytes and microglial cells, reducing both their apoptotic and inflammatory actions through multiple pathways [202,203,204,205]. Melatonin can also preserve adhesion molecules involved in tight junctions, which are necessary to maintain BBB integrity, by reducing the levels and activity of MMP-9 [196,206,207]. This action prevents the infiltration of leukocytes into the brain [208]. Lastly, by interacting with TLR4 and TLR2 and disrupting their binding to DAMPs, melatonin interferes with the activation of the inflammatory cascade (Figure 5), thereby downregulating all inflammatory signals in the post-ischemic brain tissue [4].

### 4.3. Melatonin’s Effects on Oxidative Stress

The production of RONS, which are incredibly damaging to cells due to their ability to activate inflammatory mechanisms and damage proteins and DNA, is increased during both the ischemic and reperfusion phases of an ischemic stroke. This increase is a result of mitochondrial damage caused by the aforementioned overload of Ca^2+^ inside the cells, the expression of cyclooxygenase-2 (COX-2), and the activation of nitric oxide synthases (NOS) [44,129,209,210,211,212].

Melatonin has been proven to act as a direct scavenger of free radicals, superoxide anion radicals (O_2_^•^), hydrogen peroxide (H_2_O_2_), hydroxyl radicals (^•^OH), nitric oxide (NO^•^), and peroxynitrite anions (ONOO^−^), in cells and tissues. It acts through the modulation of several signaling cascades [205,213,214,215] (Figure 5). Additionally, it acts indirectly by suppressing pro-oxidative enzymes such as COX-2 [216] and activating antioxidative enzymes [45]. Melatonin also inhibits the release of cytochrome C (an apoptotic protein) from damaged mitochondria into the cytosol [217].

### 4.4. Exploring the Neuroprotective Potential of Melatonin in Stroke: Insights into Signaling Pathways

In stroke, the excessive release of glutamate leads to the activation of NMDA, AMPA, and kainate receptors, resulting in a detrimental influx of Ca^2+^ into the cells (Figure 5). This influx ultimately triggers apoptotic cell death [218]. Melatonin acts as an inhibitor of NMDA receptors in the brain, and when combined with memantine (an NMDA receptor inhibitor), it can significantly reduce the infarct volume in an MCAO model [219].

Additionally, melatonin exerts its neuroprotective effect by modulating different signaling pathways. The heme oxygenase-1 (HO-1)/CREB signaling pathway is involved not only in the development of depression-like behaviors that can be developed after an ischemic episode, but also in the transformation of astrocytes into a neurotoxic and inflammatory type cell. Melatonin interacts with HO-1/CREB, inducing an anti-depression effect [220].

The protein kinase B (Akt) is a molecule resistant to oxidative stress in the brain, whereas sirtuin 3 (SIRT3) activates superoxide dismutase 2 (SOD2, which converts H_2_O_2_ radicals into non-toxic molecules) and modulates mitochondrial oxidative stress, thus maintaining ROS homeostasis [221]. In fact, SIRT3 expression is downregulated after an ischemic episode, which contributes to the oxidative stress that appears after it [204]. The activation of the Akt-SIRT3-SOD2 pathway by melatonin as well as the increase in SIRT3 expression alone by it are mechanisms by which melatonin can decrease infarct volume and apoptotic rate after an ischemic stroke [204,222].

One way to alleviate mitochondrial damage after an ischemic episode is by targeting mitochondrial fission. Excessive mitochondrial fission causes an increased RONS production and the liberation of pro-apoptotic factors. By stimulating the Yap–Hippo pathway (decreased in ischemia-reperfusion injuries), melatonin can promote mitochondrial fusion by enhancing the activity of optic atrophy 1 (OPA-1). This makes fragmented mitochondria interact with each other, allowing mitochondrial recovery and, in the end, decreasing the brain reperfusion stress [223].

STAT3 is one of the many transcription factors that modulate the microglia/macrophage response to tissue damage. This damage in the tissue promotes the activation of STAT3, which promotes the pro-inflammatory activation of microglial cells. Melatonin induces an anti-inflammatory effect by promoting the STAT3 phosphorylation (Figure 5), thus shifting the pro-inflammatory phenotype of microglia to an anti-inflammatory one and reducing the ischemic stroke-induced brain damage [204].

A predominant NMDA receptor 2 (NR2a) stimulation at the synaptic level promotes pro-survival signaling via activation of pro-survival proteins such as Akt, ERK, and CREB, while NR2b at extra-synaptic areas favors excitotoxic death. During ischemic conditions, NR2a levels decrease, but melatonin may exert its protective effects by attenuating the NR2a cleavage and, additionally, by increasing the γ-enolase/PI3K/AKT/CRMP2 survival pathways [224].

The antioxidant activity of melatonin involves the modulation of several signaling pathways described in detail in the next section.

## 5. Melatonin as an Antioxidant and Free Radical Scavenger in Stroke: Mechanisms and Therapeutic Implications

The generation of free radicals in stroke has significant implications for neuronal damage and the progression of ischemic injury. In this regard, the ischemic cascade triggered by stroke results in a sequence of molecular and cellular events, including the generation of RONS [225]. Thus, several mechanisms contribute to the generation of free radicals in stroke, among them (i) oxidative stress, characterized by an imbalance between the production of ROS, including O_2_^•^, H_2_O_2_, and ^•^OH, and the ability of endogenous antioxidants to neutralize them, leads to lipid peroxidation, protein oxidation, and DNA damage, which disrupt cellular function and contribute to neuronal injury and cell death [225]; (ii) mitochondrial dysfunction, leads to electron leakage from the electron transport chain, resulting in the generation of O_2_^•^ (Figure 5). Additionally, the release of mitochondrial pro-apoptotic factors during stroke contributes to the generation of ROS [226] and; (iii) the inflammatory response that occurs in stroke, is a double-edged sword, despite the fact that it plays a role in clearing debris and initiating tissue repair processes, excessive inflammation can exacerbate brain damage by promoting the release of pro-inflammatory cytokines and attracting more immune cells, such as microglia and infiltrating neutrophils, which produce RONS through the activation of NADPH oxidase and inducible nitric oxide synthase (iNOS), respectively [227]. In this context, ROS-induced oxidative stress compromises the integrity of the BBB, promoting the infiltration of immune cells and inflammatory mediators into the brain, and therefore, exacerbating neuronal damage [228].

Given all the above information, stroke is a complex condition involving multiple pathophysiological processes and molecular targets. Therefore, the exploration of novel therapeutic approaches focussing on multitarget modulation, would be a good starting point for identifying and targeting multiple critical mechanisms involved in stroke, such as inflammation and oxidative stress, among others [229]. In this regard, melatonin holds great promise in stroke management [23,230]. It is able to cross the BBB efficiently [231], ensuring effective delivery to the site of injury, and this, combined with its ability to modulate various neuroprotective pathways [232] (Figure 5), makes it an attractive therapeutic strategy. Additionally, melatonin exhibits indirect antioxidant effects by enhancing the activity of endogenous antioxidant defense systems [33]. It upregulates the expression and activity of various antioxidant enzymes, including superoxide dismutase (SOD), catalase (CAT), glutathione peroxidase (GPx), and heme oxygenase-1 (HO-1) [45,233]. These enzymes play critical roles in neutralizing RONS and detoxifying oxidative products (Figure 5). Therefore, melatonin’s ability to enhance antioxidant enzyme activity helps to restore redox balance and mitigate oxidative stress in stroke.

The antioxidant activity of melatonin against stroke involves the modulation of several signaling pathways [234]. In this context, melatonin activates the master regulator of cellular antioxidant defense mechanisms, the Nuclear Factor Erythroid 2-Related Factor 2 (Nrf2) Pathway [233], where Nrf2 translocates to the nucleus upon activation and binds to antioxidant response elements (ARE) in the DNA, leading to the upregulation of genes encoding antioxidant enzymes, such as SOD, CAT, and GPx. Activation of the Nrf2 pathway by melatonin enhances the cellular antioxidant capacity, reducing oxidative stress and protecting against stroke-induced damage [233,235] (Figure 5). Likewise, melatonin can also modulate the Kelch-like ECH-Associated Protein 1 (Keap1) pathway [236], which regulates the stability and activity of Nrf2; melatonin interacts with Keap1, preventing its inhibitory effect on Nrf2. This interaction leads to the stabilization and nuclear translocation of Nrf2, thereby increasing the expression of antioxidant enzymes and enhancing the antioxidant defenses against stroke-induced oxidative stress. Another important pathway that contributes to the neuroprotective effects of melatonin against stroke concerns protein kinase C (PKC) [237,238], which is involved in cell signaling and oxidative stress regulation. PKC activation by melatonin triggers downstream signaling events that promote antioxidant effects, such as the upregulation of antioxidant enzymes and the inhibition of pro-oxidant enzymes. PKC-mediated melatonin signaling in the Mitogen-Activated Protein Kinase (MAPK) pathway, including extracellular signal-regulated kinase (ERK), c-Jun N-terminal kinase (JNK), and p38, is involved in cellular responses to oxidative stress [239,240,241,242]. Activation of ERK and other MAPKs by melatonin [242] leads to the phosphorylation and activation of transcription factors that regulate antioxidant enzyme expression, such as activator protein 1 (AP-1) and cAMP response element-binding protein (CREB) [243]. Therefore, by modulating MAPK signaling, melatonin enhances antioxidant defenses and protects against oxidative damage in stroke. The phosphoinositide 3-kinase (PI3K)/Akt pathway, known for its role in cell survival and protection against ischemic injury, also contributes to melatonin’s antioxidant effects. Activation of the PI3K/Akt pathway by melatonin [244] leads to the phosphorylation and activation of Akt, which in turn phosphorylates and inhibits pro-apoptotic proteins and transcription factors involved in oxidative stress-induced cell death (Figure 5). By promoting cell survival and inhibiting oxidative stress-induced apoptosis, melatonin’s activation of the PI3K/Akt pathway contributes to its antioxidant effects in stroke [244,245]. Moreover, the silent information regulator 1 (SIRT1) pathway, is a NAD^+^-dependent protein deacetylase involved in cellular stress responses and longevity. Melatonin can activate SIRT1, which regulates various pathways involved in neuroprotection, including antioxidant defense, mitochondrial function, and anti-apoptotic processes [222] (Figure 5).

As a whole, these signaling pathways interact and crosstalk with each other, forming a complex network of antioxidant mechanisms that mediate the protective effects of melatonin against oxidative stress in stroke. Further research is needed to fully understand the intricate interplay and downstream effects of these pathways to optimize the use of melatonin as an antioxidant therapeutic strategy in stroke management.

Melatonin’s capability to scavenge free radicals is attributed to its unique chemical structure, containing electron-donating functional groups, such as indole and methoxy groups, which facilitate electron transfer and RONS quenching, which reduces oxidative (H_2_O_2_, ^•^OH, and O_2_^•^) and nitrosative (inhibiting the formation of ONOO^−^, a highly reactive and cytotoxic molecule formed from the reaction between nitric oxide and superoxide) stress [33,246]. Furthermore, the scavenger action of melatonin also involves the preservation of mitochondrial function in stroke; enhancing mitochondrial respiration, ATP production, electron transport chain efficiency, preventing mitochondrial permeability transition pore opening, and reducing ROS generation [24,247,248]. On the basis of this complex background, inflammation plays a key role in stroke pathogenesis and exacerbates oxidative stress [249,250]. Relative to this, melatonin indirectly reduces RONS production and scavenges free radicals in stroke by modulation of the inflammatory response (Figure 5), regulating the production of pro-inflammatory cytokines, such as tumor necrosis factor-alpha (TNF-α), IL-1β, and IL-6 as well as inhibiting the activation of NF-κB [251,252].

Overall, while melatonin’s direct antioxidant effects have been extensively studied [253,254,255,256], emerging research has shed light on the importance of its metabolites in mediating its antioxidative actions [256,257,258,259]. In this sense, after melatonin is metabolized in the body, it undergoes various enzymatic reactions, leading to the formation of several metabolites that act as free radical scavengers. The free radical scavenging properties exhibited by melatonin metabolites are attributed to their ability to directly interact with and neutralize RONS. Melatonin metabolites act as potent antioxidants by donating electrons or hydrogen atoms to RONS, thereby stabilizing, and neutralizing them. One of the primary metabolites of melatonin, 6-Hydroxymelatonin (6-OHM), is formed through the action of cytochrome P450 enzymes and exhibits potent antioxidative activity. It acts as a direct free radical scavenger, neutralizing a variety of RONS, including H_2_O_2_, ^•^OH, and ONOO^−^ and reducing oxidative stress [256]. Furthermore, 6-OHM can also stimulate the activity of various antioxidant enzymes, and inhibits lipid peroxidation, thereby reducing oxidative damage to brain cells. N1-Acetyl-N2-formyl-5-methoxykynuramine (AFMK) is another important melatonin metabolite generated through the action of cytochrome P450 enzymes, myelo- or heme peroxidases, or formed from the reaction between melatonin and free radicals, particularly RONS. AFMK possesses strong antioxidant properties and can efficiently scavenge a wide range of free radicals, including ^•^OH, ONOO^−^, and peroxyl radicals (ROO^•^) [260]. AFMK also exhibits anti-inflammatory effects, further contributing to its neuroprotective potential in stroke. N1-Acetyl-5-methoxykynuramine (AMK) is another metabolite formed through the enzymatic conversion of melatonin by indoleamine 2,3-dioxygenase (IDO) and is an active metabolite with antioxidant properties. AMK acts as a free radical scavenger, specifically targeting ROO^•^ [261].

These melatonin metabolites possess inherent free radical scavenging properties and contribute to the overall antioxidant effects of melatonin. They can directly neutralize a wide range of RONS, thereby reducing oxidative stress and protecting against oxidative damage in stroke and other oxidative stress-related conditions. However, further research is needed to fully elucidate the specific roles and mechanisms of these melatonin metabolites in neuroprotection and stroke therapy.

Several preclinical studies have investigated the potential neuroprotective effects of melatonin in stroke models [222,244,262,263,264]. These studies have consistently demonstrated the ability of melatonin to reduce infarct size, improve neurological outcomes, and attenuate oxidative stress-related markers. Furthermore, melatonin has shown the potential to enhance cerebral blood flow [265] and protect against reperfusion injury [266,267], a critical component of stroke pathogenesis. However, although the preclinical data are promising, clinical trials exploring the efficacy of melatonin in stroke are relatively limited. Some early-phase clinical studies have reported positive outcomes [268], including improved functional recovery and reduced markers of oxidative stress in patients treated with melatonin following ischemia and reperfusion injury [269]. However, larger randomized controlled trials are needed to further validate these findings and determine the optimal dosing, timing, and long-term benefits of melatonin treatment in stroke patients. If proven successful, melatonin could revolutionize stroke treatment by providing a novel and cost-effective approach to reduce neuronal damage and improve functional outcomes.

## 6. Melatonin Improved Cognitive and Behavioral Performance

The circadian rhythm of endogenous melatonin is thought to be important in maintaining the normal sleep–wake rhythm [270,271]. The sleep–wake rhythm plays an important role in the development and maintenance of cognitive functions such as memory consolidation and learning. Similarly, sleep disturbances in mild to moderate Alzheimer’s disease (AD) are associated with increased memory and cognitive impairments [272]. Melatonin secretion by the pineal gland gradually decreases with age, in circadian disorders, but especially in stress-related pathologies that trigger degenerative diseases. Degenerative diseases are the result of a continuous process of dysfunction of cells affecting tissues or organs that progressively deteriorate over time. These diseases include neurodegenerative diseases such as Parkinson’s and Alzheimer’s disease, cardiovascular diseases, metabolic syndromes such as type 2 diabetes, and neoplastic pathologies such as cancer [273]. In addition, clinical trials of melatonin have been reported for all the disorders previously mentioned [274].

The role of melatonin on cognitive function has been evaluated in different situations; for example, increased sleep efficiency and total sleep time was successfully improved, without any effect on cognitive function in breast cancer patients [275], and cognitive deficits induced by benzodiazepine treatment does not seem affected by melatonin in patients with severe mental illness [276]. Prolonged release of melatonin has positive effects on cognitive functioning and sleep maintenance in AD patients, particularly in those with insomnia comorbidity, suggesting a causal link between poor sleep and cognitive decline [277].

Melatonin exerts biological processes via activation of two G-protein-coupled receptors, and non-receptors mediated pathways [278]. Two receptors named MT1 and MT2 are preferentially coupled to Gi proteins. MT1, and most likely MT2, is also coupled to Gq/11 proteins [279,280]. The third type of receptor MT3 is an enzyme quinone reductase 2 [281] and one nuclear receptor [282]. Some of the non-receptor activities highlight the property as a free radical scavenger [283,284], with special activity in mitochondria [248] and preserving the membrane fluidity, protecting against free radicals [285], and also attenuating the laser radiation-induced mutation and deletion of mitochondrial DNA [286]. Other activity includes a direct protein interaction inhibiting the calcium-calmodulin-dependent kinase II (CaMKII) activity [287].

The rhythmic release of melatonin from the pineal gland is coordinated with the local release of melatonin in the retina and suprachiasmatic nucleus via the activation of two G protein-coupled receptors, MT1 and MT2 [288]. Melatonin concentrations in cerebrospinal fluid (CSF) is higher compared to peripheral blood, and melatonin in CSF exhibits a concentration gradient in sheep [289,290] and humans [291]. The highest concentration is measured in the third ventricle near the gland pineal recess. Thereafter, melatonin concentrations gradually decrease in CSF sampled from the middle of the third ventricle, the aqueduct, the fourth ventricle, and the lumbar subarachnoid space [292].

Different animal models with different pathologies show cognitive impairment, and several behavior protocols are used to evaluate the role of melatonin in improving this function. The Morris water maze (MWM) is a robust and reliable spatial learning and memory test and is correlated with hippocampal synaptic plasticity and NMDA receptor function [293]. The novel object recognition test (NORT) is another behavioral paradigm to evaluate recognition memory [294], and the Y-maze protocol is used to evaluate spatial working and reference memory [295]. The MWM was widely used to evaluate cognitive function, and melatonin shows an improvement in learning and memory in this behavioral test in a cognitive impairment induced by liver fibrosis in rats [296], obese mice model [297], type 2 diabetic mice model [298], cortical compact impact model [299], and repeated mild traumatic brain injury if melatonin is administered during early pathological stages but not in late (chronic) stages [300]. Induced cognitive impairment shows learning and memory improvement by melatonin in MWM, NORT, and the Y-maze behavioral tests. In contrast, intact rats show learning and memory impairment induced by melatonin treatment in the MWM test [301].

The evidence of melatonin receptors in the hippocampus [302,303] suggests the role of melatonin in the modulation of learning and memory processes. Electrophysiological studies have demonstrated the role of melatonin in modulating synaptic transmission [302,303,304] and regulation of the short-term plasticity evaluated by the paired-pulse stimulation paradigm [305]. Results exploring the excitability indicate a regulation of the firing in hippocampal neurons [302,306] through the activation of melatonin receptors. Nevertheless, melatonin reduces the long-term potentiation (LTP) in the hippocampus CA1 [307], and this LTP inhibition correlates with an impairment in cognitive behavior [301]. In agreement with behavioral data, inhibition of the LTP induction by melatonin and cognitive impairment was observed in intact subjects; whereas, the early administration of melatonin in the traumatic brain injury (TBI) model shows an increase in the frequency of spontaneous excitatory and inhibitory postsynaptic currents without affecting the frequency of the miniature postsynaptic currents in the prefrontal cortex and hippocampus. These changes in synaptic transmission were associated with an improvement of the learning and memory in the MWM behavioral test. In summary, melatonin shows beneficial effects, improving cognitive functions associated with pathological processes such as oxidative stress, neuroinflammation, excitotoxicity, or scavenging free radicals as part of the mechanisms involved in neurodegeneration, traumatic brain injury, and vascular or metabolic diseases.

## 7. Suppressing Inflammatory Response by Melatonin: Effects on the Inflammasome

As previously mentioned, the pathophysiology of ischemic brain damage involves the activation of many deleterious signaling cascades, including ionic imbalances, excessive release of glutamate [308], inflammation, and free radical-induced oxidative and nitrosative stress [309]. Oxidative stress induces the release of DAMPs that trigger an inflammatory response, resulting in microglial activation, and increased BBB permeability that leads to peripheral immune cell infiltration [310,311]. Accumulating evidence suggests that post-ischemic inflammation is responsible for the secondary progression of brain injury and that stroke severity depends on the magnitude of this inflammatory response [312,313]. One of the most potent pro-inflammatory cytokines is IL-1β, which plays an important role in ischemic stroke through the following mechanisms: (i) increased microvascular permeability, BBB dysfunction and vasogenic cerebral edema [314]; (ii) exacerbation of the inflammatory response leading to secondary brain damage by secreting and releasing various neurotoxic substances, such as IL-1β, IL-6, TNF-α, and promoting the activation of iNOS [315]; and (iii) promoting apoptosis of injured cells by activating apoptotic molecular machinery, gradually transforming the original ischemic perimeter into the cerebral infarct area, ultimately exacerbating brain injury [316].

NOD-like receptor family pyrin domain containing 3 (NLRP3) inflammasome is the most widely characterized inflammasome that activates caspase-1 which contributes to the maturation and secretion of the cytokines IL-18 and IL-1β and induces pyroptosis cell death [317]. NLRP3 inflammasome is a key step in the innate immune response. Recently, the NLRP3 inflammasome was shown to play an important role in renal, myocardial, hepatic, and cerebral ischemic stroke [318]. Inhibition of NLRP3 inflammasome activation has been proposed as a promising new therapeutic target for inflammation-related diseases. Canonical activation of the NLRP3 inflammasome requires two sequential signals, priming and activation. The priming signal is initiated through the TLR4 receptors by DAMPs or pathogen-associated molecular patterns (PAMPs) such as LPS, leading to transcription of NLRP3 inflammatory components via NF-κB activation and nuclear translocation. Activation signals are stimulations of NLRP3 by various recognized factors, leading to procaspase-1 cleavage [319]. Active caspase-1 cleaves the pro-inflammatory cytokines IL-1β, IL-18, and the protein gasdermin D into their mature forms. Finally, mature gasdermin D forms pores in the cell membrane, causing pyroptotic cell death and cytokine release [320]. IL-1β and IL-18 enhance inflammation and promote immune cell infiltration.

Melatonin has been shown to inhibit NLRP3 inflammasome activation by modulating multiple proteins and signaling pathways. First, as mentioned above, RONS are key triggers for the activation of the NLRP3 inflammasome. Thioredoxin-interacting protein (TXNIP) is one of the most important redox regulators. By inhibiting thioredoxins, TXNIP regulates the intracellular redox balance. In a model of LPS-induced endometritis, melatonin downregulates TXNIP-mediated activation of NLRP3 inflammasome [321]. Furthermore, melatonin reduces TXNIP levels in cadmium-treated primary hepatocytes and suppresses RONS production and NLRP3 activity [322]. Another signaling pathway that melatonin can regulate is Nrf2. Under oxidative stress conditions, Nrf2 translocates to the nucleus and plays a role in protecting cells from oxidative damage, leading to increased expression of antioxidant and detoxifying enzymes, thus reducing oxidative stress [323]. It has been shown that melatonin provides protection against NLRP3 inflammasome activity through the removal and clearance of Nrf2-mediated RONS [203,233].

Autophagy is known to be a negative regulator of NLRP3 activation. Mitophagy is a subtype of autophagy that helps clear dysfunctional mitochondria, and in this sense, it has been shown that melatonin increased the expression of the autophagy markers Atg 5 and LC3-II/LC3-I and the mitophagy markers PINK-1 and Parkin and suppresses NLRP3 inflammasome activation in a model of subarachnoid hemorrhage [324]. Moreover, LPS-induced inflammation leads to an increase in NLRP3 inflammasome protein due to an autophagic flux impairment, and melatonin has a potent anti-inflammatory effect by restoring the LPS-induced autophagic flux blockage, hence reducing NLRP3 inflammasome components expression and decrease in ROS production [325]. Furthermore, melatonin may inhibit multiple transcription factors associated with the priming step of NLRP3 inflammasome activation. Thus, melatonin inhibits NF-κB signaling through silent information regulator 1 (SIRT1)-dependent deacetylation of NF-κB and retinoid orphan receptor α (RORα), thereby blocking septic responses induced by cecal ligation and puncture [326]. SIRT1 is an NAD^+^-dependent deacetylase and a potent regulator of intracellular inflammatory, metabolic, and oxidative stressors. Therefore, it has been shown that SIRT1 reduces NLRP3 inflammasome activation by deacetylating NLRP3 protein, and melatonin may increase SIRT1 activity and thus inhibit NLRP3 inflammasome activation in different inflammatory models [327,328]. Finally, the regulatory function of melatonin on NLRP3 inflammasome also occurs through post-transcriptional modifications since melatonin is able to inhibit the formation of NLRP3 inflammasome complex by altering the expression of various miRNAs [327]. These are, therefore, the main mechanisms by which melatonin reduces NLRP3 inflammasome activation.

## 8. Melatonin, iNOS, and Calcium/Calmodulin

As previously mentioned, melatonin is considered a multitasking molecule, but some physiological roles have yet to be established. Melatonin is a highly lipophilic molecule that crosses easily through cellular membranes and produces its cellular functions through diverse mechanisms: first, by its interaction with specific G-protein coupled receptors identified as the MT1 and MT2 leading down-stream second messengers, activation of signaling pathways, and gene transcription [246,329,330,331]; second, by acting as a free radical scavenger detoxifying RONS [246,254]; or third, through its binding to nuclear proteins [332,333,334] and intracellular proteins such as calmodulin (CaM), regulating intracellular Ca^2+^ signaling pathways because CaM is an essential mediator of Ca^2+^ signaling within cells [335,336,337,338] (Figure 5).

CaM has been defined as one melatonin-binding protein with considerable physiological significance depending on its affinity. In 1993, Benitez-King et al. described that melatonin binds to a single site on the CaM molecule with high affinity, in a saturable, reversible, Ca^2+^-dependent, and ligand-selective manner (Kd of 188 pM and a total binding capacity Bmax of 35 pM/µ of CaM) [336]. However, other investigations indicated a much lower affinity between melatonin and CaM [339]. The controversy remains open whether melatonin effects related to CaM are direct or because of indirect intracellular signaling mechanisms. Melatonin cascades may be sometimes crosslinked, therefore protein kinase or a transcription factor may be activated by several mechanisms. The precise interconnections between melatonin and CaM widely remain to be clarified [340]. It has been reported that melatonin increased the total cellular level and synthesis of CaM [335,337]. Additionally, melatonin directly binds to antagonize CaM [335,336,341]. Activated CaM acts both directly by interaction with key target enzymes, and structural proteins, and indirectly via specific protein kinases such as CaMKII [337].

CaMKII is a family of protein kinases encoded by four genes (α, β, γ, and δ) that mediates different physiological responses triggered by increased [Ca^2+^]_i_, by its autophosphorylation, and by the binding of Ca^2+^/CaM complex on its target proteins producing phosphorylation of serine and threonine residues [246,342]. CaMKII has a dumbbell shape connected by seven-turn alpha helixes and two hydrophobic clefts with four Ca^2+^-binding domains with typical EF-hand conformation [343]. Each isoform of CaMKII is composed of three domains: (i) the association domain located at the C-terminal region, (ii) the catalytic domain located at the N-terminal region, and (iii) the regulatory domain located between association and catalytic domains [342]. The Ca^2+^/CaM complex binds to the 293–310 AA sequence in the regulatory domain to disrupt the interaction between the regulatory and the catalytic domain [344]. After its activation, the enzyme undergoes autophosphorylation at the threonine-286 residue, changing its activity to a Ca^2+^/CaM-independent manner [345,346]. Therefore, the activated Ca^2+^/CaM complex can interact with other targets such as NOS [345].

In rat hippocampal neurons, it has been shown that CaMKII activates and translocates from the cytoplasm to the presynaptic active zone [345,347] where it regulates neurotransmitter synthesis and release through phosphorylation of different targets such as K^+^ channels [348] or synaptotagmin and synapsin I [347]; additionally, CaMKII translocates to the post-synaptic density [345,349] during neuronal activation where nNOS is mainly located, therefore an interaction between both molecules occurs [345].

It has been stated that the regulation of CaMKII activity depends on its relative concentration and the concentration of its activator Ca^2+^/CaM complex [345,350]. Importantly, melatonin upregulates CaM synthesis; therefore, an increase in the activation of CaMKII is associated with melatonin. A direct effect of melatonin on PKC or its activation by diacylglycerol phosphorylates is the augmented CaM, activating CaMKII which autophosphorylates to maintain its activity-inducing dendrite growth and arborization, essential for synaptogenesis to reestablish the synaptic connectivity lost in aging neuropsychiatric disorders [338], indicating that melatonin can be used to prevent neuronal dysfunction.

As previously indicated, melatonin synthesis is upregulated under circumstances where free-radical generation is exaggerated [246,340], because it has protective actions against free-radical species acting through anti-oxidative and pro-oxidative enzymes [246,351,352] including eosinophil peroxidase, myeloperoxidase (MPO), and NOS [351,353,354,355].

It is well known that in aging the progressive accumulation of oxidative debris promotes the functional inefficiency of cellular processes inducing free-radical generation, the oxidative damage provokes apoptosis, and this loss of cells contributes to age-related deterioration. Additionally, the increased age leads to a gradually diminished melatonin production such that, in the elderly, the nocturnal melatonin rise in the circulation is greatly attenuated; this situation can have health consequences [246]. Moreover, the maintenance and progression of several diseases (neurodegenerative, cardiovascular, skin deterioration, cancer, metabolic syndrome, among others) where oxidative stress has a great role, can be accelerated by reduced production of melatonin during aging [246], supporting that melatonin is an important molecule to prevent age-related deterioration. In this sense, as reported in the previous sections, the antioxidant effect of melatonin occurs because of its ability to scavenge ROS such as hypochlorous acid (HOCl), H_2_O_2_, ^•^OH, ONOO^−^, and O_2_^•^ [353,356,357]. This occurs due to the effect on NOS activity. The link between heme destruction and disturbance of the zinc-tetrathiolate center of inducible NOS (iNOS) was described recently by Camp, leading to iNOS monomerization, protein unfolding, and accumulation of toxic free iron that occurs in many inflammatory diseases. Importantly, these events can be prevented in the presence of melatonin [353]. In this regard, iNOS expressed by immunoactive cytokines generated in tissues at sites of inflammation or in response to viral infection produces NO that is connected with essential functions for the pathogenesis of several diseases including diabetes, multiple sclerosis, and cancer, among others [358,359]. In animals, iNOS from cytokine-stimulated macrophage cell lines is a zinc homodimeric enzyme consisting of two identical subunits, containing a reductase and an oxygenase domain [353,359,360,361]. The reductase domain, C-terminus, binds NADPH, flavin mononucleotide (FMN), and flavin adenine dinucleotide (FAD), at various sites and importantly, contains a CaM binding linker peptide that tightly binds CaM [359]. The oxygenase domain (N-terminal portion) binds the iron protoporphyrin IX (heme) prosthetic group, the substrate L-arginine (L-Arg), the cofactor tetrahydrobiopterin (H4B), and a zinc atom [359]. The zinc atom is coordinated symmetrically by two cysteine residues from each subunit in a tetrahedral arrangement at the bottom of the dimer complex [361]. This complex plays an important role in maintaining the dimeric iNOS stability and preventing its monomerization [353].

The Ca^2+^/CaM binding events allow cross-subunit electron transfer from the FMN to the heme and is the essential electron transfer step because it enables O_2_ to bind to the NOS and thus start the process of NO biosynthesis [362,363]. In the brain, NO participates in numerous functions, including neurotransmission, development, and neuroprotection [364]. However, NO reacts with O_2_^•^ to form ONOO^−^, a more powerful oxidant that acts as a toxicant molecule [365] that causes lipid peroxidation, protein nitration and direct DNA damage leading to cell death [345,366]. NO is a small molecule synthesized by NOS named neuronal NOS (nNOS) and endothelial (eNOS), which are constitutive isoforms, and the iNOS that is negligible in resting cells but is induced by inflammatory cytokines and LPS [358,359,362,367,368]. LPS binds to the TLR4 on the surface of macrophage membranes, leading to the activation of MAPK or NF-κB signaling pathways and further iNOS gene expression [369] to produce NO. Additionally, microglia can exhibit a pro-inflammatory phenotype by LPS contributing to the activation of iNOS. Thus, during infection, which is more related to aging, microbial LPS can induce a prolonged low-grade inflammatory state where macrophage and microglial iNOS is upregulated [370,371].

Increased vulnerability to infection and the development of inflammatory diseases is more frequent during aging, and the onset and progression of some diseases such as neurodegeneration, cancer, metabolic syndrome, multiple sclerosis, septic shock, and adjuvant arthritis [360,372] are strongly associated with changes in the immune system function, and the loss of immune responses has been associated with resistance to bacterial infection [370]. Within the immune system the release of cytokines and chemokines is regulated by circadian phases [371,373] to maintain homeostasis through a well-organized sequence of immune defensive responses against pathogens. The decline of diurnally rhythmic immune responses related to aging amplifies disease-causing inflammation [370]. This situation could be associated with the reduced circadian production of melatonin during aging [246]. The principal biological modulators of the mammalian immune response are the macrophages population, which is heterogeneous but predominantly composed of mononuclear leukocytes [367]. Macrophages are classified as classically-activated M1 or alternatively-activated M2 [374]. M1 macrophages are pro-inflammatory cells responsible for the initiation of the immune response, while M2 macrophages are anti-inflammatory cells. M1 macrophages are activated by the microbial LPS to induce the release of pro-inflammatory interleukins IL1β, Il-6, IL-12, IL-23, and TNF-α, chemokines, hydrolyzed proteases interferons, and ROS that are released downstream in the innate immunity response [375]; this activation promotes the increase in NO-mediated by iNOS activation.

On the other hand, microglia are cells involved in regulatory processes critical for development, maintenance of the neural environment, response to injury and subsequent repair, and sense pathological events in the CNS to orchestrate innate immune responses. They are regulated by the CNS microenvironment and are the first line of defense against invading microbes and via interactions with neurons can be the first to detect critical changes in neuronal activity and health [376]. With aging, a decrease in synaptic plasticity is accompanied by the ability of microglia to express pro-inflammatory cytokines to contribute to mild chronic inflammatory conditions, also accompanied by a decrease in anti-inflammatory cytokines [376] and an increase in NO synthesis. In this sense, melatonin increased the amount of CaM and phosphorylation of CaMKII prompting dendritogenesis in rat hippocampal slices suggesting that melatonin could repair the loss of synaptic connectivity in aging neuropsychiatric disorders [338]. Additionally, melatonin has been associated with neurogenesis stimulation and neuronal survival [377,378], this adds evidence for its potential use as an adjuvant in neurodegenerative diseases

The effects of melatonin in immunoregulation have been well-described [367,379,380]. Melatonin shows an inhibitory effect against the LPS effects only if melatonin is administered before LPS in J774.2 macrophages. This effect occurs because melatonin reduces iNOS steady-state mRNA levels and iNOS protein expression, and it was associated with inhibition of the NFκ-B. Inhibition of iNOS-derived NO production contributes to the anti-inflammatory effect produced by melatonin [356]. Additionally, the inhibitory effect of melatonin on iNOS is due to an interaction with CaM [79], and iNOS is associated with CaM in a tightly bound form [358]. In the immunostimulated macrophages, prolonged exposure to melatonin during the induction of iNOS and its association with CaM increases the inhibitory effect on iNOS activity [356]. Melatonin exerts a potent anti-inflammatory effect and reduces NO production in murine models of carrageenan-induced inflammation [381]. Protection by melatonin in shock and inflammation is due to the inhibition of iNOS expression [246,356]. However as described previously, the reduced melatonin production by aging could be a factor that contributes to low-grade inflammatory conditions.

On another hand, visfatin an adipokine highly expressed in adipocytes and macrophages [382] exerts an inflammatory activity through the expression of pro-inflammatory cytokines (TNF-α, IL-6, and IL-1β) in different cell types [382,383,384]. In RAW 264.7 macrophages, melatonin reduces the visfatin-induced iNOS expression through the suppression of NF-κB in a similar way as LPS-induced iNOS expression. Visfatin may contribute to the enhancement of obesity-associated inflammation via the release from macrophages. Interestingly visfatin has been found to increase in human dental pulps with age [385], contributing to senescence, and is highly expressed in inflammatory diseases and some cancers [384]; however, the functional role of visfatin in aging has not yet been well established, but the protective melatonin effect against visfatin-induced inflammation through iNOS activation, strongly supports its use to treat the low-grade inflammatory condition.

## 9. Melatonin in Microglial and Astrocyte Cell Activation

The beneficial effects of melatonin have been shown in pre-clinical and/or clinical studies for several neurodegenerative conditions caused by insults such as infections, hypoxia/ischemia, trauma, or toxins as well as for neurodegenerative diseases, such as Alzheimer’s disease (AD), Parkinson’s disease (PD), multiple sclerosis (MS), and amyotrophic lateral sclerosis (ALS) [12,221,258,386,387,388,389,390,391,392,393]. A shared feature among these pathologies is neuroinflammation, a complex process mediated primarily by microglia and astrocytes. In this regard, recent studies have revealed the potential use of melatonin as a neuroprotective agent, especially in the context of glial cell activation.

### 9.1. Melatonin in Astrocyte Activation

Astrocytes, the most abundant non-neuronal cells in the brain, were classically considered to serve passive structural and support roles in the brain. They are now considered full-fledged participants in brain circuitry and processing, with a wide range of functions at the cellular level. These functions include the formation, maturation, and elimination of synapses, maintaining the integrity of the BBB and ionic homeostasis, clearing neurotransmitters, regulating the volume of the extracellular space, and modulating synaptic activity and plasticity [394,395,396]. Moreover, astrocytes play an essential role in the initiation, execution, and regulation of immune responses in the CNS [397,398,399].

Astrocytes undergo morphological, molecular, and functional remodeling, known as astrocyte activation, in response to various pathological stimuli, including inflammation, tumors, trauma, ischemia, epilepsy, and neurodegeneration [400,401,402]. Astrocyte activation leads to the loss of their normal homeostatic functions and the acquisition of protective or detrimental roles, including proliferation, scar-border formation, immune cell recruitment, and neurotoxicity [400,403,404,405]. Just as the M1/M2 activation axis has been described for microglia, there has been a proposal for astrocyte A1/A2 phenotypic polarization. According to this model, A1 denotes reactive cells that have lost their neurosupportive functions and have become toxic toward neurons, while A2 refers to inflammation-resolving astrocytes [406,407]. However, this simplistic binary classification statement to characterize the reactive states of astrocytes is under debate [400].

Following an ischemic stroke, the peri-infarct region is commonly divided into two parts: an inner region adjacent to the lesion with a high density of monocytes/macrophages and an outer region rich in astrocytes [408]. Reactive astrocytes in the outer region secrete various pro-inflammatory cytokines, chemokines, and matrix metalloproteinases, which disrupt the BBB and attract peripheral leukocytes, exerting deleterious effects on the brain after ischemia [408,409]. Additionally, astrocytes in this region display elongated and polarized processes with an increased GFAP expression and expression of factors involved in extracellular matrix remodeling and clustering of reactive astrocytes [410]. After ischemia and reperfusion injury, melatonin has been shown to attenuate reactive astrogliosis and glial scar formation, reduce the infarct volume, and, consequently, enhance axonal regeneration and promote neurobehavioral recovery in adult rats [411,412]. These effects of melatonin are related to a reduction in the activity of glycogen synthase kinase-3 beta (GSK-3β) and receptor-interacting serine/threonine-protein 1 kinase (RIP1K), proteins involved in astrocyte responses, after the treatment [412]. Furthermore, melatonin pretreatment has been shown to reduce the increased expression of Nox2 and Nox4 in astrocyte neurons and endothelial cells, reduce RONS levels, and inhibit cell apoptosis [413].

Traumatic brain injury (TBI) also induces reactive astrogliosis accompanied by an increased expression in intermediate filaments such as glial fibrillary acidic protein (GFAP) and vimentin, along with astrocyte hypertrophy and functional alterations as glutamate and potassium clearance changes [414]. Melatonin has been shown to partially reverse TBI-induced anxiety-like behavior in rats and decrease the number of activated astrocytes and neuronal apoptosis in the amygdala induced by TBI [415]. Melatonin also has been demonstrated to decrease the number of A1-type astrocytes in the NAc after TBI and mitigate TBI-induced depression by activating of HO-1/CREB signaling [220]. Similar results have been obtained in the hippocampus, dentate gyrus [300], and cortex [300,416] where melatonin reduces activated astrocytes and cytokine production and promotes cell survival and cognitive function after TBI induction. Furthermore, melatonin treatment suppresses the accumulation and the proliferation of microglia and astrocytes, down-regulates caspase-3, Bax and GFAP expressions, and the pro-inflammatory markers iNOS, IL-1β, and TNF-α expressions in a spinal cord injury model [417].

Astrocytes play a crucial role in the development of neurological and neurodegenerative diseases, primarily due to the disruption of normal homeostatic function and the acquisition of toxic functions [418]. Additionally, the accumulation of certain protein aggregates such as alpha-synuclein (SNCA), Aβ, and Tau in astrocytic cytoplasm can contribute to the pathology of these diseases [419], representing a significant characteristic in various neurodegenerative conditions. Furthermore, astrocytes can undergo various detrimental transformations, resulting in either atrophy and loss of function or reactive astrogliosis with hypertrophy [420]. Although melatonin has been shown to have a beneficial role in different neurodegenerative diseases such as AD [421,422], PD [389,423], MS [424], or ALS [425], less is known of its effects on astrocyte activation. In the rat hippocampus, melatonin has been shown to attenuate synaptic dysfunction and reduce astrogliosis improving the Aβ1–42-induced impairment in spatial learning and memory [426]. On the other hand, in results obtained by Andrade et al., melatonin reduces Aβ levels but does not reduce GFAP levels in rats receiving ICV-STZ (intracerebroventricular streptozotocin STZ) [427]. In an MS model induced by cuprizone, melatonin also reduces astrocyte activation in young and aging mice due to the antiapoptotic, antioxidant, anti-inflammatory, and neurotrophic effects [428]. Melatonin has also been demonstrated to prevent the development of neuropathic pain in the cuneate nucleus in a lysophosphatidylcholine (LPC)-induced median nerve demyelination neuropathy model. Here, melatonin reduces astrogliosis and through MT2 receptors inhibits the activation of astrocytic MAPKs, production of pro-inflammatory cytokines, and development of demyelination-induced neuropathic pain behavior.

Some neurotoxins and drugs also produce neuroinflammation and neuronal cell death. Melatonin also has been demonstrated to be effective in decreasing astrogliosis and the production of pro-inflammatory cytokines in these situations. In trimethyltin chloride- treated mice, melatonin reduces the expression of C3, Gbp2, and Serping1, indicating the suppression of A1 reactive astrocytes in the brain [429]. Furthermore, melatonin has inhibitory effects on caspase-3 and GFAP up-regulation induced by the plasticizer Diisononyl phthalate [430]. Additionally, melatonin reduced reactive astrocytes associated with the activation of the TLR4/MyD88/NFκB signaling pathway by methamphetamine [384,431].

### 9.2. Melatonin in Microglia Activation

The diverse functions of microglia play a crucial role in the onset, progression, and resolution of inflammation within the CNS. These processes also involve communication and interaction between microglia and various other cell types, particularly astrocytes and neurons, although it is not limited to them. Classically, phenotyping macrophages and microglia (cells that are functionally and developmentally related) into M1 (resting) and M2 (activated) based on the expression of markers related to these categories, was used to indirectly assume a detrimental M1 or beneficial M2 microglial role. In this classic description, M1 microglia releases inflammatory mediators and induce inflammation and neurotoxicity, while M2 microglia release anti-inflammatory mediators and induce anti-inflammatory and neuroprotection. However, this dualistic classification of good or bad microglia is very controversial and is inconsistent with the wide repertoire of microglial states and functions in development, plasticity, aging, and diseases that were elucidated in recent years (see [432] for more information).

As previously mentioned, melatonin can suppress proinflammatory signals and has been extensively documented. Notably, research has focused on its impact on iNOS and COX-2, as well as its ability to inhibit inflammasome activation, particularly NLRP3. Moreover, it also activates processes in an anti-inflammatory network, in which SIRT1 activation, upregulation of Nrf2, downregulation of NF-κB, and release of the anti-inflammatory cytokines IL-4 and IL-10 are involved [258,433,434]. However, these findings often did not specifically focus on microglia.

Microglia and infiltrated macrophages initially polarize toward a neuroprotective anti-inflammatory phenotype after stroke, but gradually transform into a detrimental pro-inflammatory phenotype [435,436]. Melatonin treatment ameliorates brain damage after ischemic stroke at least partially through shifting microglia phenotype from pro-inflammatory to anti-inflammatory polarity [204,411,437,438,439]. This M2 phenotype promoting effect by melatonin is via signal transducer and activator of transcription 1/6 and 3 (STAT1/6 and 3) and neuronal melatonin type 1 receptor activation [204,411,439]. After TBI, melatonin also suppresses microglial activation and the production of proinflammatory cytokines reducing mTOR pathway phosphorylation [440,441]. Similar effects of melatonin have been shown in the spinal cord injury (SCI) model. Melatonin decreases the expression levels of M1 microglia phenotypic markers (CD16, iNOS, and TNF-α) and increases M2 markers (Arg1, CD206, and TGF-β) also reducing the levels of pro-inflammatory cytokines (TNF-α, IL-6, and IL-1β) in the SCI mice and rats, facilitating functional recovery [417,442].

Both M1 and M2 microglial phenotypes are involved in the pathogenesis of neurodegenerative diseases [443], and promoting microglia polarization shift from M1 to M2 phenotype may be a prospective strategy in the therapy of neurodegenerative diseases such as AD, PD, ALS, and MS [444]. Melatonin has been shown to ameliorate neuroinflammation by inhibiting STAT-related pro-inflammatory (M1-like) polarization of microglia in a cellular PD model [445]. Moreover, melatonin attenuates microglial activation by negatively regulating NLRP3 inflammasome activation via a sirtuin 1 (SIRT1)-dependent pathway in 1-methyl-4-phenyl-1,2,3,6-tetrahydropyridine (MPTP)-induced murine PD models [446]. Regarding AD, melatonin has been shown to promote anti-inflammatory microglial activation rescuing SIRT1 and BDNF expression/release following β-amyloid (Aβ42)-induced microglial activation [447] and remediates the cytokine profile of Tau-exposed microglia [448]. In other neurological diseases, such as epilepsy, melatonin can also promote the polarization status of microglia from an M1 to M2 phenotype. In this case, Li et al. [445] have demonstrated that melatonin plays an antiepileptic role in KA-induced temporal lobe epilepsy, reducing the frequency and severity of seizures and changing microglia polarization status by regulating the RhoA/ROCK signaling pathway.

## 10. Post-Stroke Melatonin Treatment

Currently, the only treatment for ischemic stroke, which is limited and ineffective in a large number of patients, is thrombolysis when administered within 4 h of its onset [449]. However, it is very common for stroke patients to experience multiple complications such as depression, movement disorders, epilepsy, fatigue, dysphagia, or vascular dementia [450,451,452,453,454,455]. Due to its beneficial properties and lack of toxicity even at high concentrations, melatonin could be proposed as a potential treatment for alleviating post-stroke complications, although research on this topic is limited and more research is required to determine the correct doses and timing at which melatonin may be most effective.

Central post-stroke pain (CPSP) is another issue that can arise because of a stroke. It is a neuropathic pain syndrome characterized by pain and sensory abnormalities in the body parts corresponding to the injured brain territory [456]. A study found that CPSP-model rats showed improved performance in several pain tests when treated with intraperitoneal melatonin, in a dose-dependent manner. This suggests a neuromodulatory effect of melatonin in the treatment of CPSP [263,457].

Considering melatonin’s anti-inflammatory and antioxidant properties, it could be administered as a post-stroke treatment to alleviate the sequelae caused by increased inflammation and RONS. In an aged MCAO rat model, Rancan et al. [458] demonstrated that oral melatonin administration after surgery can inhibit the upregulation of pro-apoptotic markers associated with ischemic stroke, suggesting a reduction in neuronal damage. Melatonin administration after ischemic injury can also significantly decrease the activity of mitochondrial enzymes responsible for oxidative stress, such as NOS and COX-2, thereby reducing the infarct area [459]. Additionally, in a rat model of ischemia/reperfusion injury, another study found that intranasal administration of melatonin loaded in lipidic nanocapsules post-stroke can increase the number of surviving neurons in the hippocampal CA1 region and improve oxidative stress and inflammatory marker levels in the hippocampus. These effects were even more pronounced than those observed with oral melatonin administration [460].

Additionally, although melatonin does not possess direct antiviral capacity, supplemental melatonin has shown favorable effects, eliminating the pathogenicity of highly deadly viruses [461,462,463,464,465]. This has been extensively documented in various case reports, with the prevention of hemorrhagic shock syndrome from Ebola virus infection being particularly promising [466]. Given melatonin’s pharmacological profile, which highlights its potent abilities to moderate inflammation, mitigate oxidative stress, and regulate immune response, this indoleamine should be considered a preferred candidate for testing in the context of attenuating hyper-inflammation in COVID-19 patients and enhancing the success of their clinical management.

## 11. Conclusions

The conclusions that can be drawn from this review are the following:

Melatonin as a Promising Neuroprotective Agent: Melatonin shows considerable potential beyond its role in regulating sleep–wake cycles. Its neuroprotective properties position it as a promising candidate for stroke therapy, offering protection against ischemic brain damage.

Powerful Antioxidant and Possible Anti-Aging Agent: Melatonin exhibits antioxidant effects, making it a potential anti-aging agent by mitigating oxidative stress, which is implicated in age-related neurodegenerative diseases.

Implications of Free Radicals in Stroke and Aging: Free radicals, highly reactive molecules, are of great interest in medical research due to their involvement in multiple pathologies. Free radicals play a crucial role in oxidative stress and cellular damage, which has significant implications for understanding complex biological processes such as stroke and aging.

Interaction of Melatonin, Free Radicals, and Non-Exciting Amino Acids: Both melatonin and non-excitatory amino acids play integral roles in the stroke and aging processes, with melatonin acting as an antioxidant and non-excitatory amino acids providing neuroprotection through modulation of several key factors.

Potential of Therapeutic Strategies: Understanding the interaction between melatonin, free radicals, and non-excitatory amino acids opens opportunities for therapeutic strategies in age-related diseases and stroke prevention/treatment. We highlight the need for additional research to fully understand these mechanisms and explore their therapeutic applications in future clinical trials.

Future Perspectives and Improvement of Quality of Life: Deepening our understanding of these mechanisms brings us closer to developing specific interventions that can improve neuronal resilience and quality of life in people affected by stroke and age-related neurodegenerative disorders.

## Figures and Tables

**Figure 1 antioxidants-12-01844-f001:**
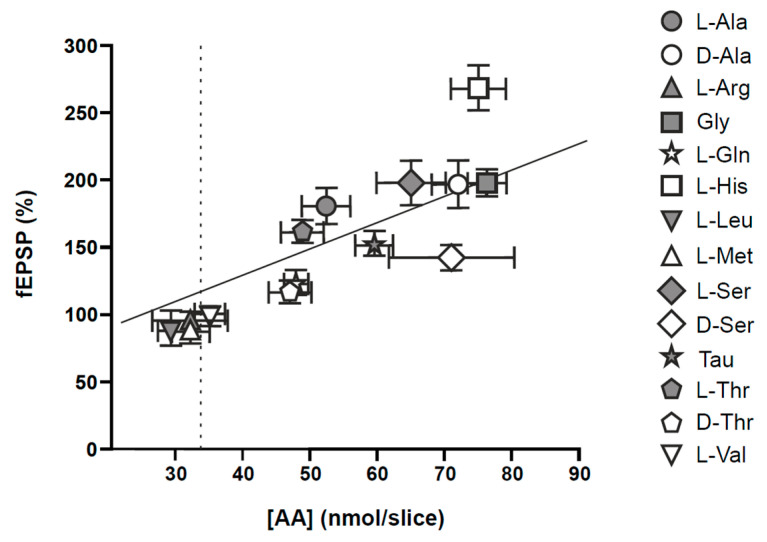
Positive correlation between fEPSP potentiation and amino acid accumulation represented by the solid straight line of linear regression (R^2^ = 0.45; *p* < 0.001). The dashed line indicates the total amino acid content of control slices. Adapted with permission from ref. [157].

**Figure 2 antioxidants-12-01844-f002:**
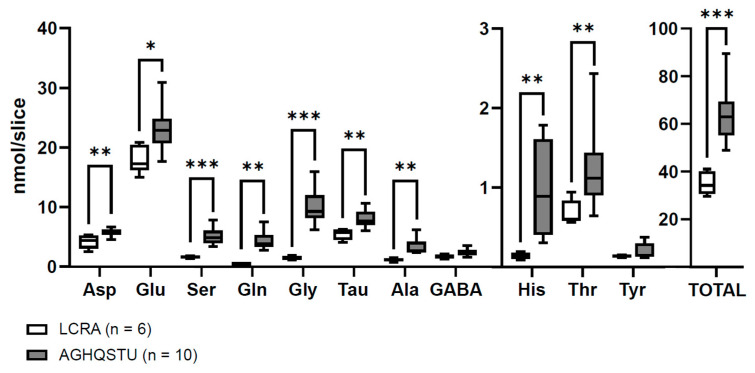
Amino acid content of control slices (white boxes) and after the application of AGHQSTU (alanine, glutamine, glycine, histidine, serine, taurine, and threonine; grey boxes). The total amino acid content has been determined as the sum of all amino acids quantified and reflected in this figure, including tyrosine. * *p* < 0.05; ** *p* < 0.01; and *** *p* < 0.001. One-way ANOVA, followed by Tukey’s test for multiple comparisons of parametric data and Kruskal–Wallis test followed by Dunn’s test for multiple comparisons of nonparametric data. The numbers in parentheses indicate the number of slices. Adapted with permission from ref. [165].

**Figure 3 antioxidants-12-01844-f003:**
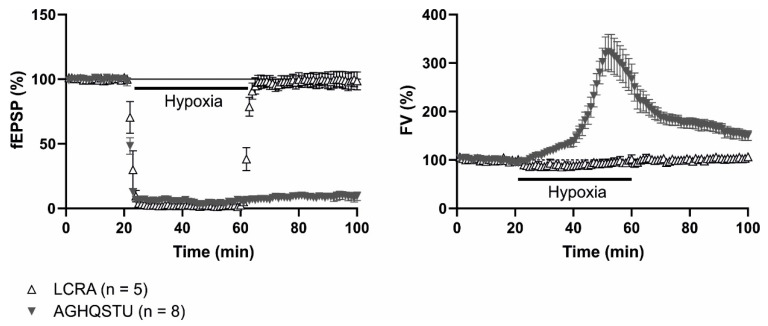
The hypoxia-induced depression of synaptic potentials becomes irreversible in the presence of AGHQSTU (alanine, glutamine, glycine, histidine, serine, taurine, and threonine; black symbols). Time course of changes in fEPSP (**left panel**) and FV (**right panel**) elicited by a 40 min period of hypoxia (indicated by the horizontal bar). LCRA: artificial cerebrospinal fluid. Adapted with permission from ref. [165].

**Figure 4 antioxidants-12-01844-f004:**
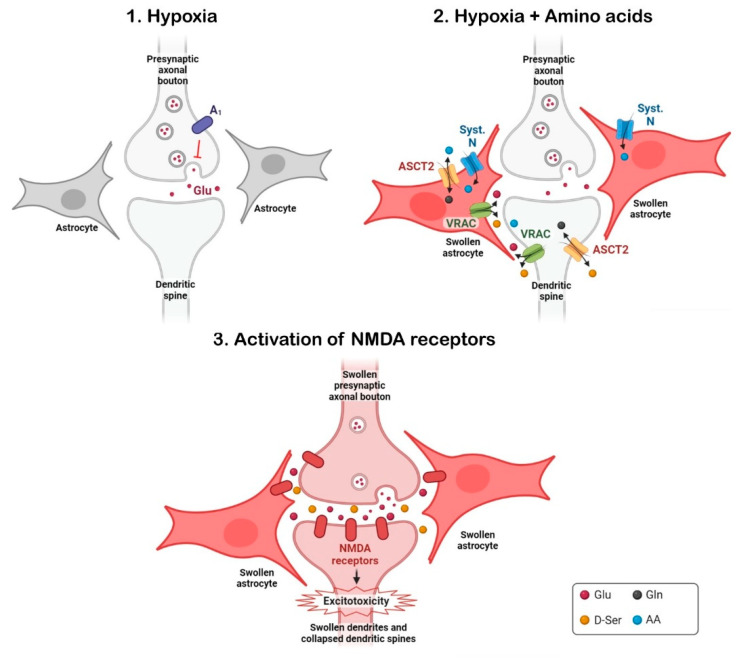
Graphical representation of the proposed model. (**1**) During hypoxia, adenosine release inhibits glutamatergic transmission via its A1 receptors. (**2**) Uptake of non-excitatory amino acids by glial cells (via transporters such as those of the N-system and ASCT2) leads to an increase in cell volume. The activation of volume-regulated anion channels (VRAC) releases excitotoxins (such as glutamate, aspartate, and D-serine) into the already reduced extracellular space. This glutamate is in addition to glutamate released by other pathways during ischemia (synaptic glutamate, reversal of its transporters, etc.). (**3**) Glutamate and D-serine released into the reduced interstitial space leads to a progressive increase in their extracellular concentration and the activation of NMDA receptors, triggering excitotoxicity phenomena that result in increased cell volume and dendritic beading. Glu: glutamate, Gln: glutamine, D-Ser: D-serine, AA: other amino acids. Image designed with BioRender.com.

**Figure 5 antioxidants-12-01844-f005:**
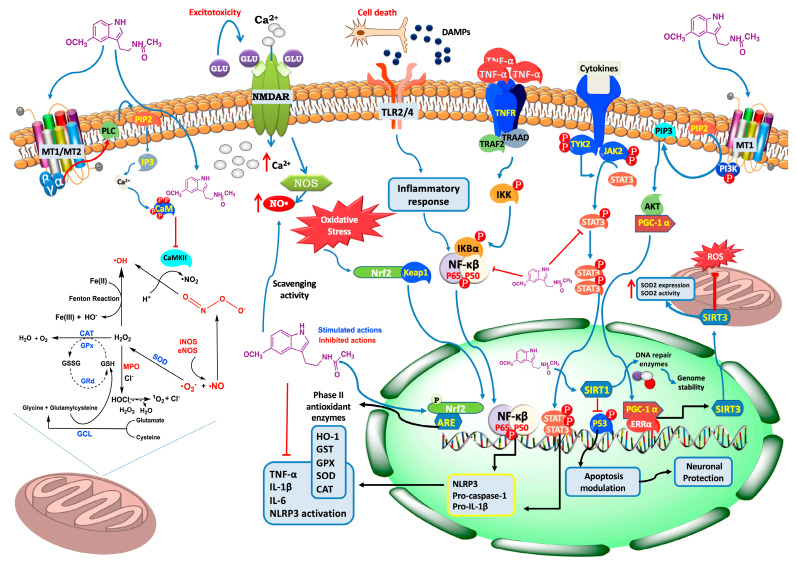
Schematic view showing melatonin-mediated actions in stroke. In neuronal cells, glutamate (GLU) interacts with the NMDA receptor (NMDAR), boosting a high intracellular Ca^2+^ and NO^•^ levels by activating nitric oxide synthase (NOS), which causes a reactive oxygen and nitrogen species (RONS) flux overflow within the cell. Subsequently, melatonin’s ability to easily diffuse allows it to enter neuronal cells, where it binds to calmodulin with high affinity inhibiting CaMKII (Ca^2+^/calmodulin dependent protein kinase-II), a major mediator of neuronal cell death involved in pathological glutamate signaling, thereby mitigating excitotoxicity damage. Furthermore, radicals are generated as a result of electron leakage from the inner mitochondrial membrane’s electron transport chain; the rogue electrons chemically convert adjacent oxygen molecules to produce the superoxide anion radical (O_2_^•^). This reactant either combines with nitric oxide to form the highly oxidizing peroxynitrite anion (ONOO^−^) or is immediately dismutated by superoxide dismutase 2 (SOD2) to hydrogen peroxide (H_2_O_2_). Both the Fenton reaction and the kinetically sluggish Haber–Weiss reaction require a transition metal, such as ferrous iron (Fe^2+^), to convert H_2_O_2_ to the hydroxyl radical (^•^OH). The ^•^OH damages molecules along with other oxidants, which promotes neuronal cell death. In this regard, melatonin acts as a direct radical scavenger (directly neutralizes ^•^OH and the ONOO^−^), regulating RONS overload, as well as an indirect agent, boosting antioxidant enzymes such as glutathione peroxidase (GPx) and SOD2, preserving mitochondrial homeostasis and, as a result, enhancing cellular energy efficiency. In addition, melatonin attenuates oxidative stress by stimulating Nuclear erythroid-related factor 2 (Nrf2), the principal transcription factor that regulates antioxidant response element (ARE)-mediated expression of phase II detoxifying antioxidant enzymes. Under normal conditions, Nrf2 is sequestered in the cytoplasm by an actin-binding (Kelch-like) protein (Keap1); on exposure of cells to oxidative stress, Nrf2 dissociates from Keap1, translocates into the nucleus, binds to ARE, and transactivates phase II detoxifying and antioxidant genes. Among the spectrum of antioxidant genes controlled by Nrf2 are catalase, SOD, hemoxigenase-1 (HO-1), and GPx. Melatonin increases Nrf2 gene expression. Melatonin also functions as an anti-inflammatory. Thus, DAMPs, released from damaged or dying cells, promote pathological inflammatory response through toll-like receptors (TLR2/4). In this context, TNF-α acts by binding to its receptor TNFR (TNF receptor), which recruits TRADD (TNF Receptor-Associated Death Domain). This protein binds to TRAF2 (TNF Receptor-Associated Factor-2) to phosphorylate and activate the IKK (I-KappaB-Alpha kinase complex). Then, the IKK complex phosphorylates IKBα, resulting in the translocation of NF-κβ (Nuclear Factor-κβ) to the nucleus where it targets many coding genes for mediators of inflammatory responses. Subsequently, in response to cytokine receptor stimulation STAT3 (Signal Transducers and Activators of Transcription) following tyrosine and JAK2 (Janus-family tyrosine kinases) phosphorylation, dimerize and translocate to the nucleus. In addition, melatonin reverts these pro-inflammatory effects by inhibiting the JAK2/STAT3 signaling pathway and NF-κβ translocation. Additionally, melatonin-mediated signaling through MT1 receptors promotes PI3K/Akt phosphorylation. Akt coactivator-1-α (PGC-1α) complex dimerize and translocate to the nucleus where estrogen-related receptor-α (ERRα) codes to sirtuin 3 (SIRT3) gene, upregulating the expression and activity of SOD2. In this regard, melatonin is able to activate the multifaceted regulator SIRT1, which enhances the stress response by repressing p53 and promoting DNA repair. Stimulation (blue colored) or inhibition (red colored) by melatonin are also shown.

## Data Availability

No new data were created or analyzed in this study. Data sharing is not applicable to this article.

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
