# Peer review of "Non-Excitatory Amino Acids, Melatonin, and Free Radicals: Examining the Role in Stroke and Aging"

_antioxidants, 2023, doi:10.3390/antiox12101844_

Round 1

Reviewer 1 Report

The authors submitted an interesting paper for review summarizing the current state of knowledge on the role of melatonin, free radicals and amino acids in the aging process and the risk of stroke. a very interesting manuscript, but it is worth making a few corrections to make it interesting for a wider audience, including clinicians.
- most processes concern not only metabolic changes but also local blood flow mechanisms. Please add a subsection (maybe in the introduction) regarding this issue and the mediators involved in these processes. Please pay attention to nitric oxide and carbon monoxide as the youngest class of gaseous mediators in these processes.
- what is the role of processes related to the guanylate cyclase pathway. Its role in regulating blood flow is enormous, as well as in the processes leading to the progression of heart failure or pulmonary arterial hypertension. What is the interference of melatonin and the described compounds in this pathway?
- regulation of flow may depend on modulation of the vascular reaction through three types of contraction, i.e. it may occur at different stages of the receptor-effector chain. please add a few sentences about the participation of metatonin and the described compounds in these processes.
- one example of the above may be mastoparan-7, which activates the G protein. Works describing the reactivity of small resistance vessels (e.g. in the model of the rat tail artery), due to the nature of the tissue response, indicate the possibility of direct action as well as influence through the production of free radicals.
- authors of experimental works often "forget" about a very important aspect or it is not clearly presented in the manuscript. These are the concentrations of the tested compounds, but in relation to the physiological situation or disease. Based on such work, we obtain information about the effects of certain compounds, but in concentrations sometimes hundreds of times higher than the actual ones. Please try to add information about the concentration ranges of compounds used in experiments in the cited works in relation to human physiology (or pathology). Are they even comparable?
- finally, a note about the clinic. When the authors use the term stroke, they seem to ignore the etiology. over 75% of strokes are caused by a thromboembolic mechanism. Of the remaining cases, after extended diagnostics, it turns out that half of them are caused by this mechanism. The indispensable treatment is the use of DOACs. We have already published observational studies assessing 8-year concentrations of these drugs. Can they influence the production of free radicals? Please pay attention to rivaroxaban and apixaban. Could this interference be subject to a synergistic effect with melatonin and the other compounds evaluated? What is the risk of bleeding complications then?

As I mentioned at the beginning, the review is very interesting and I expect that after its update it will be considered for publication.

Reviewer 2 Report

Jimenez Carretero and co-authors prepared an informative review manuscript on convergent signaling pathways among non-excitatory amino acids, melatonin, and free radicals in the brain with implications for aging and stroke. I only have one minor suggestion for improvement below.

(1) To be consistent with the rest of the primary text of the manuscript, adjust "Hipoxia" indicated in Figure 3 to read "Hypoxia".

Author Response

Reviewer 2 Comments for the Author...

… I only have one minor suggestion for improvement below....

Thank you very much.

Minor suggestion:

  • To be consistent with the rest of the primary text of the manuscript, adjust "Hipoxia" indicated in Figure 3 to read "Hypoxia".

The indicated typographical error has been corrected.

Reviewer 3 Report

The review explores the relationship between these factors and their role in stroke and aging. Melatonin has diverse physiological functions and potential therapeutic benefits by reducing oxidative stress, inflammation, and apoptosis. Non-excitatory amino acids have neuroprotective properties, including antioxidant and anti-inflammatory effects. Free radicals cause cellular damage and contribute to age-related decline. The review also describes the pathophysiology of ischemic stroke, the most common cerebrovascular disease and the leading cause of disability worldwide. Stroke is associated with aging, which is the most significant risk factor. Aging also causes structural and functional impairments in the neurovascular unit, exacerbating the risk and severity of ischemic stroke. The authors discussed the effects of non-excitatory amino acids, such as alanine, glutamine, glycine, histidine, serine, taurine, and threonine, on synaptic transmission and cellular swelling during hypoxia-ischemia episodes. The intracellular accumulation of these amino acids could activate NMDA receptors and cause excitotoxicity. The review summarizes the multiple mechanisms by which melatonin exerts neuroprotection in the context of ischemic stroke. Melatonin can reduce cerebral edema by preserving blood-brain barrier integrity and regulating aquaporin-4 expression. Melatonin can modulate the post-stroke inflammatory response by downregulating NF-κB, TLR4, cytokines, and MMP-9. Melatonin can also scavenge free radicals and enhance antioxidant enzymes, reducing oxidative stress and mitochondrial damage. Specific comments:

1.          The introduction section does not provide a transparent background and rationale for the review. It should explain why melatonin, free radicals, and non-excitatory amino acids are relevant to stroke and aging and what are the current knowledge gaps and controversies in this field. It should also state the specific aims and scope of the review.

2.          The iconographies and tables are particularly welcome for the review article to attract the readers. However, if the authors do not hold the copyright of the figures, authors should seek the appropriate permission from the copyright holder.

3.          The review does not provide a comprehensive and critical analysis of the existing literature. It should summarize, compare, and contrast the findings from different studies and discuss their strengths, limitations, and implications. It should also identify the gaps in knowledge and the challenges.

4.          The review does not provide a clear and concise conclusion. It should summarize the study’s main findings and implications, highlight the current research’s limitations and challenges, and suggest directions for future research.

Round 2

Reviewer 1 Report

The authors have significantly modified the manuscript and it may now be considered for publication.